# Cleavage-intermediate Lassa virus trimer elicits neutralizing responses, identifies neutralizing nanobodies, and reveals an apex-situated site-of-vulnerability

Jason Gorman [1], Crystal Sao-Fong Cheung[1,5], Zhijian Duan[2,5], Li Ou[1,5], Maple Wang[3,5], Xuejun Chen [1], Cheng Cheng[1], Andrea Biju[1], Yaping Sun[2], Pengfei Wang [3], Yongping Yang[1], Baoshan Zhang[1], Jeffrey C. Boyington[1], Tatsiana Bylund[1], Sam Charaf[1], Steven J. Chen[1], Haijuan Du[1], Amy R. Henry[1], Tracy Liu[1], Edward K. Sarfo [1], Chaim A. Schramm [1], Chen-Hsiang Shen[1], Tyler Stephens[4], I-Ting Teng[1], John-Paul Todd[1], Yaroslav Tsybovsky [4], Raffaello Verardi [1], Danyi Wang [1], Shuishu Wang [1], Zhantong Wang[1], Cheng-Yan Zheng[1], Tongqing Zhou [1], Daniel C. Douek [1], John R. Mascola[1], David D. Ho[3] ✉, Mitchell Ho [2] ✉ & Peter D. Kwong [1] ✉

Lassa virus (LASV) infection is expanding outside its traditionally endemic areas in West Africa, posing a pandemic biothreat. LASV-neutralizing antibodies, moreover, have proven difficult to elicit. To gain insight into LASV neutralization, here we develop a prefusion-stabilized LASV glycoprotein trimer (GPC), pan it against phage libraries comprising single-domain antibodies (nanobodies) from shark and camel, and identify one, D5, which neutralizes LASV. Cryo-EM analyses reveal D5 to recognize a cleavage-dependent site-of-vulnerability at the trimer apex. The recognized site appears specific to GPC intermediates, with protomers lacking full cleavage between GP1 and GP2 subunits. Guinea pig immunizations with the prefusion-stabilized cleavage-intermediate LASV GPC, first as trimer and then as a nanoparticle, induce neutralizing responses, targeting multiple epitopes including that of D5; we identify a neutralizing antibody (GP23) from the immunized guinea pigs. Collectively, our findings define a prefusion-stabilized GPC trimer, reveal an apex-situated site-of-vulnerability, and demonstrate elicitation of LASV-neutralizing responses by a cleavage-intermediate LASV trimer.

Lassa virus (LASV) is an enveloped RNA virus, an Old World arenavirus, which causes Lassa fever, an acute viral hemorrhagic illness; Lassa fever is prevalent in West Africa, affecting 100,000 to 300,000 of individuals each year (https://www.cdc.gov/vhf/lassa/index.html) and causing ~5000 deaths[1,2]. To date, no licensed vaccine is available for the prevention of Lassa fever, and the only treatment is Ribavirin, a broad-spectrum antiviral drug[3]. LASV infection of humans generally occurs via contact with urine or feces of infected rodents; human-to-human transmission can then occur via direct contact[4–6]. LASV has been included on the priority pathogen list for WHO's R&D Blueprint for Action to Prevent Epidemics in an urgent effort to develop effective vaccines[7,8].

The surface of LASV virions is covered by the trimeric type 1-fusion glycoprotein complex (GPC)[9,10], the only antigen available for virus-neutralizing antibodies. Each protomer of the GPC comprises a receptor-binding GP1 subunit, a transmembrane-spanning GP2 subunit, and a stably associated signal peptide[11–15]. The LASV GPC has proven difficult to produce. Diskin and colleagues[15] have succeeded in expressing small quantities of a detergent-solubilized LASV GPC, which they subjected to cryogenic electron microscopy (cryo-EM) analysis, which revealed the signal peptide to associate as an additional transmembrane spanning segment and the matriglycan receptor to bind at the trimer apex. Saphire and colleagues[11,16] have created a higher expressing disulfide-linking GP1 and GP2 subunits, which - when bound by a quaternary specific antibody - forms soluble GPCs recognized by most neutralizing antibodies. This construct, GPCysR4, was modified to substitute the native SKI-1 cleavage site (RRLL) with a site amenable to furin cleavage. In addition, Saphire and colleagues also created another construct, GPCysRRLL, as it has been shown that the RRLL cleavage site is important for receptor binding[15] and recognition of neutralizing antibody 8.9F[17]. Furthermore, Brouwer and colleagues have shown that appending the I53-50A trimerization domain[18] to the GPCysR4 construct enables the formation of soluble GPCs, in the absence of binding antibody (Supplementary Table 1).

Although highly glycosylated GPC can induce potent T and B cell immune responses, the development of neutralizing antibodies in response to GPC is often weak and inconsistent[18–23]. From analysis of 17 convalescent subjects, 16 neutralizing antibodies have been identified that fall into four competition groups: GP1-A, GPC-A, GPC-B, and GPC-C, based on their recognition and cross-reactivity[24]. Recently, Brouwer et al.[18] have shown LASV GPC-I53-50 nanoparticles – while not able to elicit neutralizing responses in guinea pigs – could elicit neutralizing responses in rabbits (ID$_{50}$ titers of ~100) and have even identified a rabbit neutralizing antibody (LAVA01).

To provide insight into LASV GPC requirements for antibody-mediated neutralization, we designed a stable soluble GPC trimer based on the GPCysR4 structure (PDB: 5VK2)[11], by appending the T4-phage foldon trimerization domain and engineering an inter-protomer disulfide. We verified the prefusion conformation of this stabilized GPC trimer by antigenic and cryo-EM studies and showed that, when used as an immunogen – first as a soluble trimer and then as a GPC nanoparticle – we could elicit Lassa virus-neutralizing responses in guinea pigs, and we isolated a Lassa virus-neutralizing antibody, GP23, from immunized guinea pigs. We also used phage display to identify single domain antibodies (also called nanobodies) that neutralized the Lassa virus and determined the cryo-EM structure of the most potent neutralizing nanobody. The revealed nanobody epitope, at the trimer apex, appeared to require a lack of full cleavage between GP1 and GP2 subunits. Altogether, our results highlight the development of a soluble cleavage-intermediate LASV trimer, identifies nanobodies capable of neutralizing LASV, and reveals an apex-situated site-of-vulnerability as a vaccine target. We propose this apex-situated site as a target for LASV vaccine and therapeutic development.

## Results

### Structure-based design and characterization of prefusion-stabilized LASV GPC

Like other type I-viral fusion machines, the LASV GPC refolds from a metastable prefusion conformation to a more stable post-fusion conformation during cell entry[9,25]. Vaccine designers have sought to express prefusion-stabilized GPC trimers, as this form of the GPC is generally best recognized by neutralizing antibodies[11,16,26]. But which prefusion form of the GPC trimer is the optimal immunogen? While the detergent-solubilized GPC described by Diskin and colleagues appears to be a very good mimic of the virion associated mature trimer, the quantities that have been generated have been too minute to enable immunzation[15]. By contrast, the GPCysR4 with appended I53-50A

trimerization domain does not bind matriglycan and does not have an associated signal peptide, but it is able to bind most neutralizing antibodies and can be expressed in suitable quantities for immunization (Supplementary Table 1).

Starting with the GPCysR4 structure of the antibody-bound prefusion LASV GPC (PDB ID: 5VK2), we designed over 150 variants (Supplementary Data 1), which we screened for binding to the quaternary-specific GPC antibody 37.7H (from GPC-B competition group). From the screening results (Supplementary Fig. 1), we identified an engineered inter-protomer disulfide bond that links GP1 subunit of one protomer to the GP2 subunit of a neighboring protomer to yield improved antigenicity. This inter-protomer disulfide, R207GC-L326C, replaced the engineered intra-protomer disulfide C207-C360 present in the GPCysR4 construct, by reverting the residue at position 360 to a Gly and introducing a L326C mutation. R207 was replaced by Gly-Cys (R207GC) to allow the two Cys side chains to have optimal geometry for the formation of R207GC$_{GP1}$-L326C$_{GP2}$ inter-protomer disulfide bond (Fig. 1a). To further stabilize the trimeric conformation of the LASV GPC, we appended a T4 phage-fibritin (foldon) trimerization domain at the C-terminus of GP2 soluble construct (Fig. 1a).

The resultant LASV GPC trimer expressed as a soluble protein with a final yield of ~0.5 mg/L by transient transfection of mammalian cells (Supplementary Fig. 2). The purified protein yielded a major band at an expected size of a trimer (~200 kDa) on SDS-PAGE in the absence of reducing agent, indicative of the formation of an inter-protomer disulfide bond (Fig. 1b). In the presence of reducing agent, bands for GP1 and GP2 appeared; however, the majority of the stabilized trimer existed as a band of the size for a protomer, indicating that GP1 and GP2 subunits were inefficiently cleaved (Fig. 1b). To verify the prefusion conformation of the stabilized LASV GPC trimer, we analyzed its antigenicity by bio-layer interferometry (BLI) for recognition by a panel of 10 human LASV neutralizing antibodies from the four competition groups (Supplementary Table 2). The GP1-A group comprises three antibodies (10.4B, 12.1 F, and 19.7E) that bind GP1 but not GP2; the other three groups recognize only the fully assembled GPC. GPC-A group comprises three antibodies: 8.11 G, 25.10 C, and 36.1 F; GPC-B comprises antibodies 18.5 C, 25.6 A, and 37.7H; and GPC-C comprises a single antibody: 8.9 F. Of these, all except for the GPC-C antibody 8.9 F could bind, suggesting that the stabilized trimer possessed generally similar antigenic property as the GPCysR4 construct[11] (Fig. 1c). Negative-stain electron microscopy (EM) confirmed the homogeneous size and shape expected for a stable trimer in prefusion conformation of this stabilized GPC (Fig. 1d). More importantly, in the presence of various human neutralizing Fabs, the stabilized trimer preserved its trimeric shape (Fig. 1d). To further examine the overall architecture of the inter-protomer disulfide-stabilized LASV GPC trimer, we determined a cryo-EM map at 5.8 Å resolution from 47,597 particles (Fig. 1e). We observed trimers with C3 symmetry, but most of the particles displayed C1 symmetry. Lastly, our stabilized GPC trimer could withstand physical stress under various temperature (50–90 °C), pH (3.5 and 10), osmolarity (10 mM and 3000 mM NaCl), and freeze-thaw conditions, as evidenced by the retained 37.7H reactivity[11] after treatment (Fig. 1f). Overall, our design yielded a soluble prefusion-stabilized LASV GPC, which exhibited desired antigenic and structural characteristics.

### Potent neutralizing responses elicited by Lassa GPC trimer and nanoparticle immunizations

To assess the ability of the prefusion-stabilized LASV GPC to elicit neutralizing responses, we immunized guinea pigs six times with GPC trimer, followed by seven times with GPC trimer in nanoparticle format, which we made by appending a C-terminal SpyTag to the GPC trimer and coupling to encapsulin (Enc) nanoparticles that displayed SpyCatcher, as previously described[27] (Fig. 2a–c). ELISA titers to GPC

trimer saturated after three GPC trimer immunizations; similarly, ELISA titers to encapsulin-SpyCatcher (Enc-spyC) saturated after three nanoparticle immunizations (Fig. 2d). We used the Josiah strain to assess neutralizing responses, which were mostly absence during GPC trimer immunization, but gradually rose so that by the 5th and 6th nanoparticle immunization, they averaged about 500 $ID_{50}$. To assess the breadth of the elicited neutralizing response, we further assessed neutralization at week 64 on LASV lineages, observing neutralization of the Pinneo strain that was comparable to that of the Josiah strain, but lower titers against other lineages (Supplementary Fig. 3a). Sera at week 64 also competed with binding of epitopes from five neutralizing antibodies (Supplementary Fig. 3b–d).

To compare the immunogenicity of GPC trimer and GPC trimer-nanoparticle, we performed an immunization study with rhesus macaques (Supplementary Fig. 3e). We immunized one group of five macaques with GPC trimer and a second group of five macaques with GPC trimer-nanoparticles. ELISA titers to GPC trimer saturated after two immunizations (Supplementary Fig. 3f). Two animals from the GPC trimer group and one from the nanoparticle group showed neutralization above background after three immunizations; in the GPC trimer group titers reached 1832 and 715 $ID_{50}$ after the fifth immunization, and in the nanoparticle group titer reached 636 $ID_{50}$ after the third immunization (Supplementary Fig. 3g). Overall, the results in both guinea pigs and rhesus macaques demonstrate the ability of our stabilized GPC trimers to elicit neutralizing responses.

### Identification of a Lassa-virus neutralizing antibody

To provide insight into the elicited neutralization, we screened B cells from the immunized guinea pigs for those that bound the Lassa virus GPC trimer, synthesizing antibodies from those that bound, and screening synthesized antibodies for binding and neutralization (Fig. 3a). Of the 23 synthesized antibodies that bound the Lassa virus GPC trimer (Supplementary Fig. 4), a single antibody, GP23, neutralized Lassa virus (Fig. 3b). Neutralization of the Josiah strain by GP23 was similar in potency to the 37.7H antibody (Fig. 3c), with neutralization observed with Lassa virus strains from lineages II, IV, and V. Mapping of the GP23 recognition site was achieved by cryo-EM analysis, which yielded a map with nominal resolution of 5.4 Å, though with residue-level detail hampered by Fab-Fab aggregation (Fig. 3d, Table 1). The reconstructed electron density showed the closest known epitope to that of GP23 to be the neutralizing antibody 36.1 F, with which there was considerable overlap (Fig. 3d).

### Identification of LASV GPC-directed neutralizing nanobodies

Having verified our stabilized GPC trimers could elicit neutralizing responses, we sought to use them to identify neutralizing nanobodies, as these might reveal additional sites of vulnerability. We screened phage display libraries of variable domain of New Antigen Receptors ($V_{NAR}$) from sharks[28,29] and single variable domain on heavy chain ($V_{H}H$) antibodies from camels[30,31]. After four consecutive rounds of panning, phage was enriched by 400-1200-fold for binding the stabilized LASV GPC trimer (Supplementary Fig. 5). At the end of the fourth round of panning, we identified four individual clones (B8, B10, C3 from shark $V_{NAR}$, and D5 from camel $V_{H}H$) that exhibited enhanced binding to stabilized LASV GPC trimer in ELISA, whereas no binding was observed with the control protein bovine serum albumin (Supplementary Fig. 6a). The binding affinities of these single domain antibodies to the stabilized GPC trimer were in the range of 19-44 nM as measured by BLI (Supplementary Fig. 6b, Supplementary Table S2). Binding competition revealed that nanobodies C3 and D5 competed for binding to GPC with known human LASV neutralizing antibodies from the GPC-B group but not from groups GP1-A and GPC-A; nanobodies B8, and B10 showed only mild competition with the human neutralizing antibodies for binding the GPC trimer (Supplementary Fig. 6c).

We next assessed the single domain antibodies for neutralizing activity against pseudotyped LASV Josiah strain. At the tested concentration range (1 µg/ml – 1 mg/ml), all four single domain antibodies displayed some neutralization activity but failed to reach the 50% threshold (Fig. 4a). Since improved neutralization potency with multimeric single domain antibodies have been reported for other viruses[32–35], we linked a single domain antibody molecule to a llama IgG2a hinge region and the Fc domain of human IgG1[36,37]. We found that three of the four bivalent IgG2a antibodies exhibited modest neutralization toward the Josiah pseudovirus with $IC_{50}$ values ranging from 12 to 260 µg/ml (Fig. 4b). Of these bivalent IgG2a antibodies, D5-IgG2a demonstrated the best neutralizing activity with an $IC_{50}$ of 12 µg/ml (Fig. 4b). Overall, the results indicated that D5 in bivalent IgG2a format, similar to bivalent human GP1-A and GPC-A antibodies, showed enhancement of LASV neutralization by avidity[38,39]. This enhancement was not observed for bivalent human GPC-B antibodies, in which a single Fab engages two protomers in a quaternary manner (Fig. 4c).

### Cryo-EM structure of LASV GPC in complex with nanobody D5 and 8.11 G Fab

We sought to visualize binding by the most potent of the nanobodies, D5. We were unable to interpret cryo-EM images of D5 bound to GPC, but the addition of human GPC-B antibody 8.11 G led to better resolved reconstructions, enabling determination of the cryo-EM structure at 4.7 Å of our prefusion-stabilized LASV GPC trimer in complex with a single D5 nanobody and two GPC-A 8.11 G Fabs (Fig. 5a and Supplementary Fig. 7, Table 1). Several other populations with a single D5 and zero, one, or three 8.11 G Fabs were also observed but the best overall resolution was with the two Fabs, so we focused our structural analysis on that class. D5 bound to the apex of the GPC trimer, forming asymmetric interactions with all three GPC protomers (Fig. 5b). 8.11 G bound to an interface between GP1 and GP2 subunits of single protomer. Despite the heavy glycosylation of the GPC trimer, antibody 8.11 G navigated through the glycan shield, contacting four glycans (Fig. 5a-c). The 8.11 G epitope overlapped substantially with those of GP23 and 36.1 F, although lacking complete overlap (Fig. 5d).

The structure of the trimer bound by D5 displayed an asymmetric assembly and when compared to crystallized GPCs[11,16], a single protomer aligned with a root-mean-square deviation (rmsd) of 1.5 Å after removing outliers; however, neighboring protomers extended over 17 Å farther at certain regions near the apex with overall rmsds of 13.2 Å and 21.9 Å (Fig. 6a). This extension resulted in the allosteric disruption of 37.7H binding, and stabilization of this conformation by D5 appeared to prevent 37.7H from binding, thereby providing an explanation for the observed competition between D5 and GPC-B antibodies, which bind across adjacent GPC protomers distal from the apex (Fig. 6b). The asymmetry in the protomer interface extensions was matched by the protrusion of an uncleaved furin site, between GP1 and GP2 (Fig. 6c). This resulted in two internal cleavage sites and a single protruding loop, disrupting trimer symmetry and inflating the radius of the apex cavity targeted by D5. Since D5 is capable of neutralizing LASV, this asymmetric conformation may resemble a biologically relevant GPC cleavage-intermediate (Fig. 6d).

### An apex-situated site-of-vulnerability centers on the LASV receptor-binding site for matriglycan

We evaluated the D5-defined site-of-vulnerability at the GPC trimer apex for its potential utility as a vaccine target. Residues within a 5-Å footprint of the nanobody were encompassed in all three protomers with buried surface areas of 730, 620, and 350 $Å^2$ (Fig. 7a). The asymmetrically opened conformation, defined by the binding of D5, revealed an apex site with little steric hindrance from the apex glycans (Fig. 7b). D5 was more centrally located and penetrated the apex cavity farther than the human apex-binding antibody 8.9F[17] (Supplementary

**a** Design of stabilized GPC trimer

GPC trimer with DS and foldon stabilization

Inter-protomer DS by mutation R207GC (GP1) and L326C (GP2)

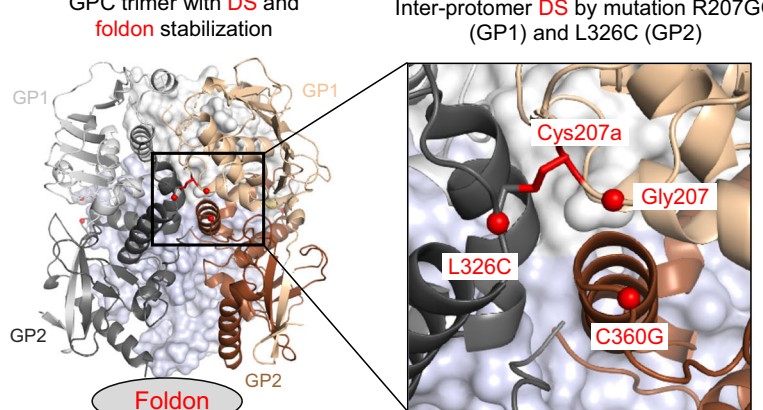

**b**

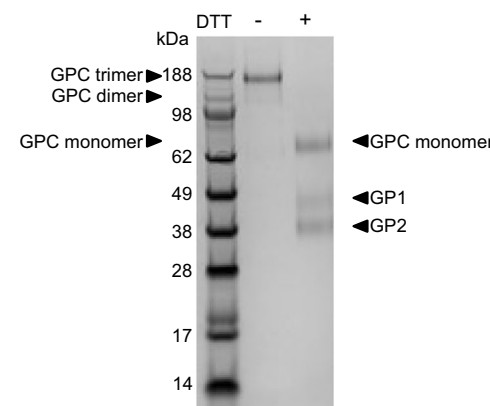

**c** Antigenicity of stabilized GPC trimer

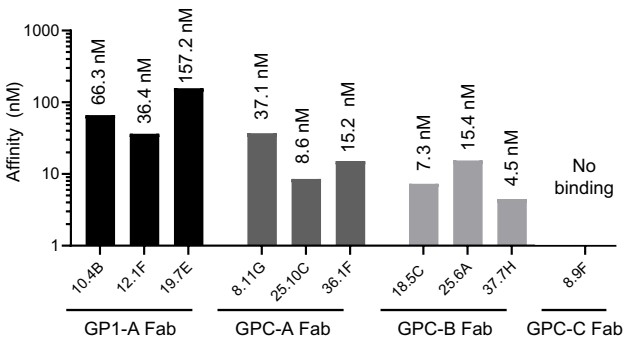

**d** EM images of stabilized GPC trimer alone and in complex with human neutralizing Fabs

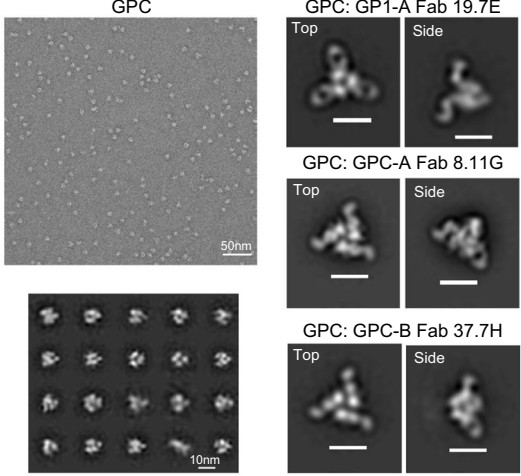

**e** Cryo-EM structure of stabilized GPC trimer at 5.8 Å

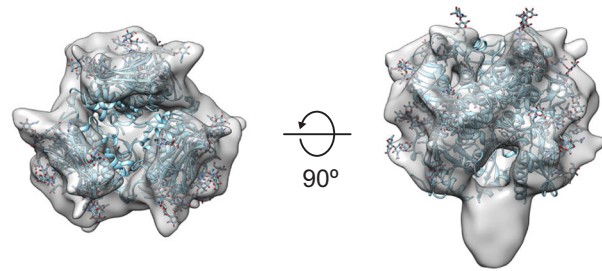

**f** Stability of stabilized GPC trimer

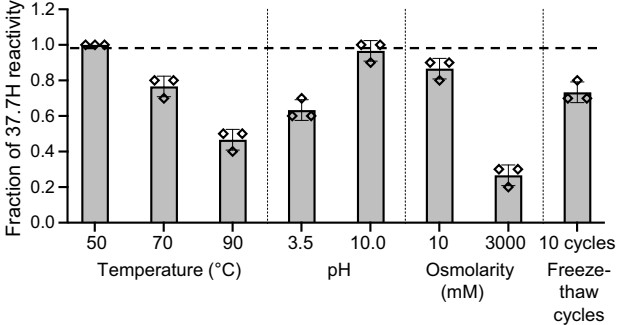

**Fig. 1 | Design and characterization of stabilized soluble Lassa GPC trimer.**
**a** Structure-based design of stabilized soluble Lassa GPC trimer. An inter-protomer disulfide (DS) bond linked GP1 of one protomer to GP2 of a neighboring protomer and a foldon domain was appended to the C-terminus of GP2. Inset shows a view of the inter-protomer DS shown as spheres. The two protomers are shown as ribbons in light gray (GP1) and dark gray (GP2), and wheat (GP1) and brown (GP2).
**b** Representative SDS-PAGE of stabilized Lassa trimer under non-reducing and reducing conditions. A high molecular weight band three times the size of the monomeric form was observed under non-reducing conditions. Three repeats show the same results. Source image is provided in the Source Data file. **c** Binding affinity of the stabilized Lassa GPC trimer towards Fabs of four groups of human Lassa-neutralizing antibodies, GP1-A, GPC-A, GPC-B, and GPC-C. **d** A representative

negative-stain EM micrograph (top left) and 2D class averages (bottom left) are shown of the stabilized Lassa GPC trimer, along with 2D class averages of the GPC trimer in complex with human neutralizing Fabs (right panels). At least 25 micrographs were recorded in each case. The 2D class averages of the complexes include representative top and side views. For right panels, the scale bars represent 10 nm. **e** Cryo-EM structure of the stabilized Lassa GPC at 5.8 Å resolution confirmed its trimeric association. **f** Physical properties of the stabilized Lassa GPC trimer. Stability of the stabilized trimer was assessed by fractional binding reactivity to antibody 37.7H after treatments under various temperatures, pHs, osmolarities, and freeze-thaw cycles. Triplicate measurements were made, and results are shown as mean ± SEM. The dotted line shows the antibody reactivity of trimer prior to physical stress.

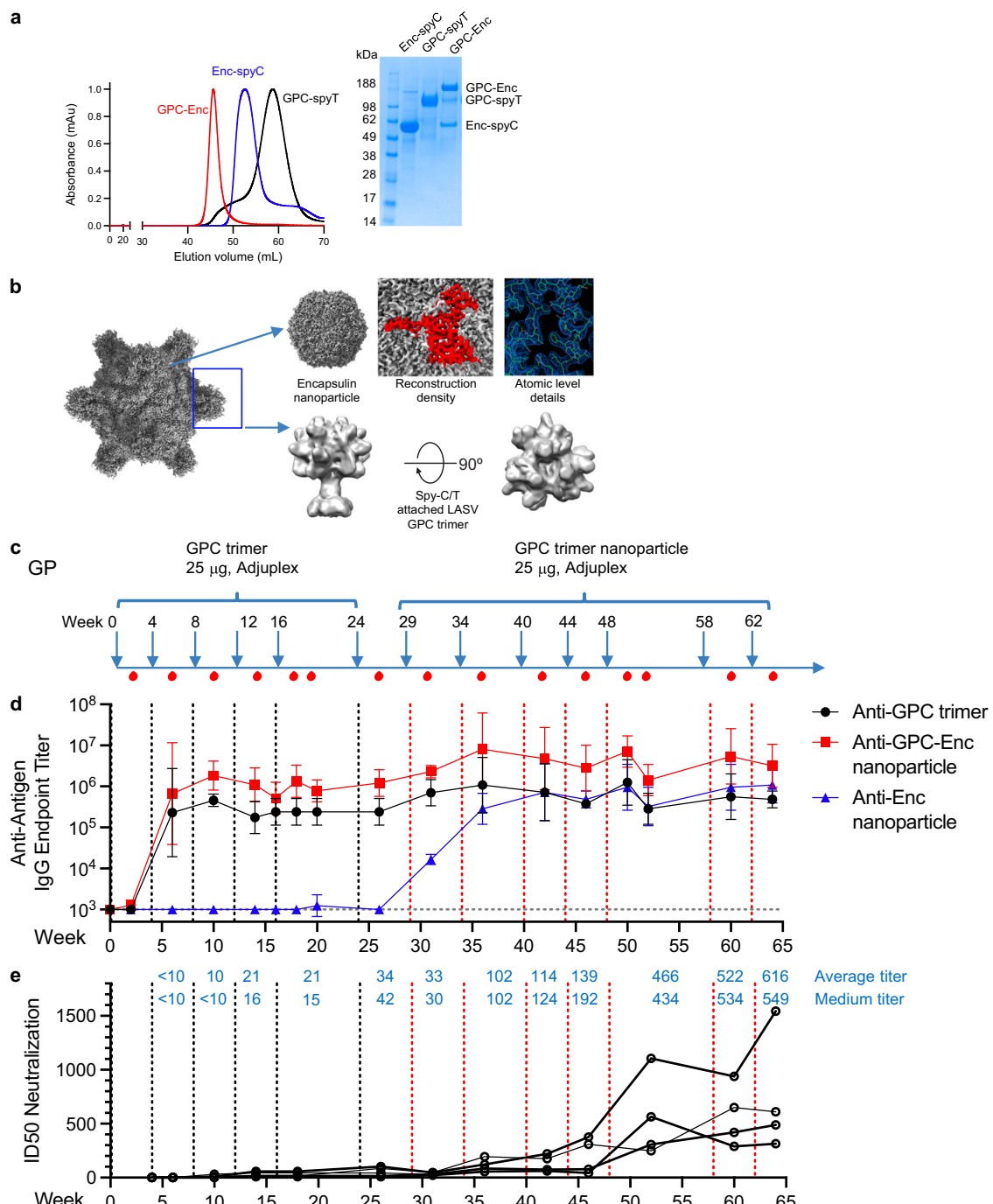

**Fig. 2 | Potent neutralizing response elicited by Lassa GPC trimer and nano-particle immunization in guinea pigs. a** Preparation of GPC nanoparticle. Lassa GPC nanoparticle profiles are shown from three individual size exclusion chromatographs. A representative SDS PAGE gel is shown from two repeats, source image is provided in the Source Data file. **b** CryoEM structure of Nanoparticle Lassa GPC-Enc. **c** Schematic representation of the immunization schedule. **d** Immune responses against to Lassa GPC trimer, GPC nanoparticle and nanoparticle alone were measured by ELISA. Initial dilution is shown as a horizontal dotted line. Data are presented as mean values +/- SEM from $n = 10$ independent biological replicates. **e** Lassa GPC trimer immunizations followed by multiple nanoparticles boosts elicited high neutralizing responses in guinea pigs. Source data are provided as a Source Data file.

Fig. 7e). Analysis of fully cleaved GPC structures[11,15,16] indicated that D5 would be obstructed from binding at the mature apex (Fig. 7c). Notably, the apex also serves as the receptor-binding site for the matriglycan of α-dystroglycan[15], suggesting D5 binding to be incompatible with matriglycan receptor binding.

Sequence analysis of representative arenaviruses displayed an expected clustering and conservation of New and Old world sequences, and we mapped this sequence variability to the surface

of the GPC structures (Fig. 7d). Greater conservation was displayed on the GP2 subunit versus the GP1 subunit. Surprisingly, residues proximal to the LASV- matriglycan receptor binding site displayed little conservation except for internal cleavage residues at the center of the apex cavity, which make pivotal contacts with matriglycan (Fig. 7d)[15]. The arenavirus family did not show strong conservation in the matriglycan-receptor binding region, which is somewhat expected given that different family members utilize

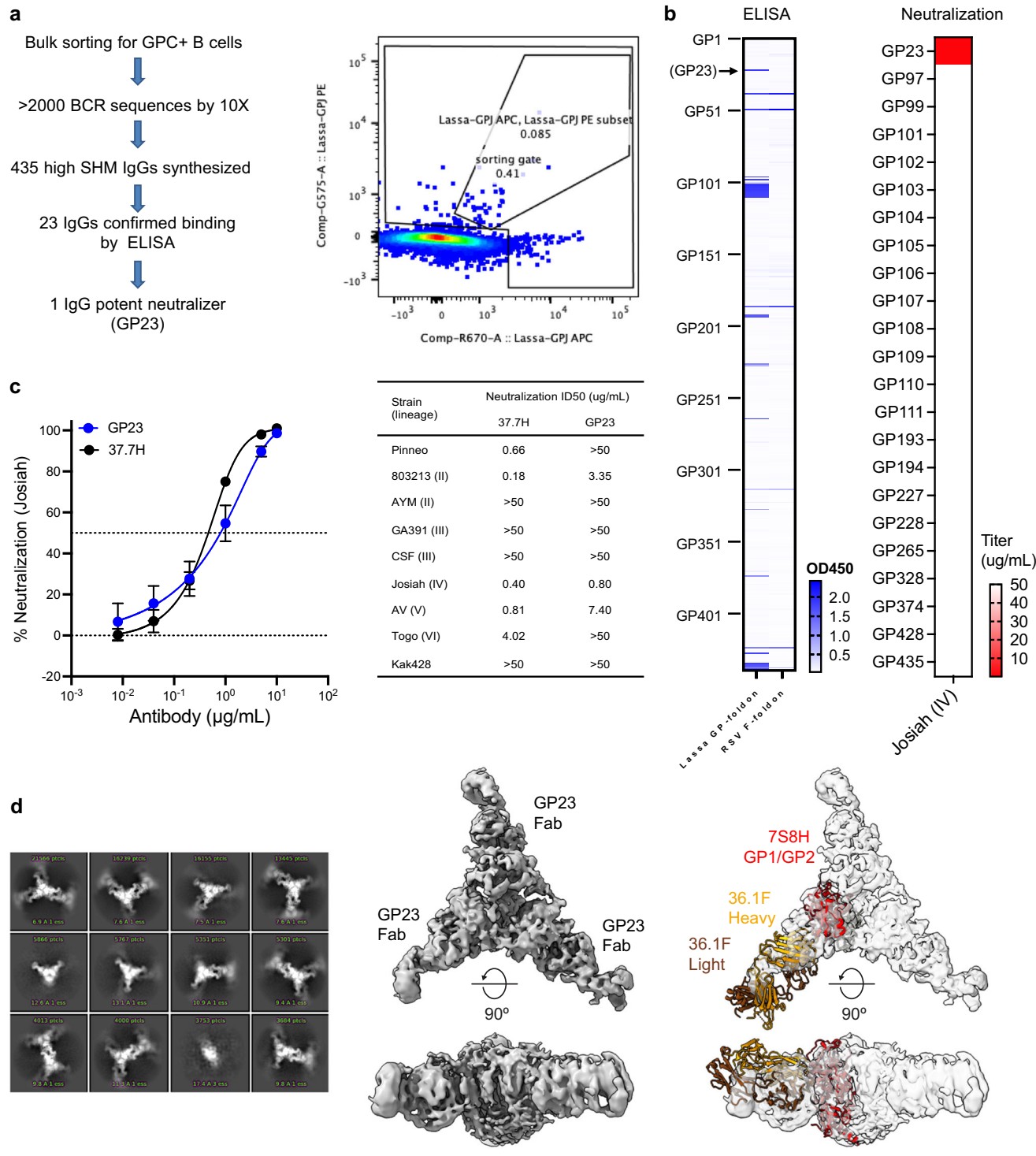

**Fig. 3 | Potent Lassa virus neutralizing antibody elicited by stabilized GPC trimer−trimer nanoparticle immunization. a** Workflow and B-cell FACS analysis for antibodies isolated from guinea pigs. **b** ELISA screening of expressed guinea pig IgGs for Lassa GPC-specific binding (left). Selected IgGs with positive ELISA binding were further assessed Lassa Josiah neutralization. **c** Neutralization on Lassa Josiah strain (left) and diverse strains (right). Neutralization was determined by interpolation after fitting data globally to a 5-parameter dose-response curve. The IC50 (dot) and 95% CI (error bar) of a global fit of six serial dilutions. **d** Cryo-EM analysis shows GP23 binding to overlap with the epitope of 36.1 F from PDB ID 7S8H. Aggregation illustrated in the 2D class images of the GP23-GPC sample (left) did not allow for accurate model building despite a nominal resolution of 5.4 Å.

different receptors. The D5 epitope was partially conserved in Old world arenaviruses but not arenaviruses in general (Fig. 7d), where the major site of conservation involved cleavage-site residues. Overall, the D5-apex site-of-vulnerability was conserved within Lassa virus lineages, but not among more diverse arenaviruses.

## Discussion

The LASV GPC trimer is metastable, conformationally labile, and heavily glycosylated and induces weak and inconsistent neutralizing responses in lassa fever survivors[40]. Stabilization by quaternary-specific human neutralizing antibodies of the GPC-B class have enabled structural analysis of the LASV GPC trimer[11,16]. Structural

**Table 1 | Cryo-EM data, reconstruction, refinement, and validation statistics**

| | Lassa GPC trimer in complex with Fab 8.11 G and nanobody D5 | Ligand-free Lassa GPC trimer C1 symmetry | Ligand-free Lassa GPC trimer C3 symmetry | Lassa GPC trimer in complex with Fab GP23 |
|---|---|---|---|---|
| EMDB ID | 41048 | 26740 | 26859 | 41302 |
| PDB ID | 8T5C | | | |
| Data collection | | | | |
| Microscope | FEI Titan Krios | FEI Titan Krios | FEI Titan Krios | FEI Titan Krios |
| Voltage (kV) | 300 | 300 | 300 | 300 |
| Electron dose ($e^-/Å^2$) | 51.15 | 56.52 | 56.52 | 51.37 |
| Detector | Gatan K3 | Gatan K2 | Gatan K2 | Gatan K3 |
| Pixel Size (Å) | 1.083 | 1.076 | 1.076 | 1.083 |
| Defocus Range (μm) | −1.0 to −2.5 | −1.0 to −2.5 | −1.0 to −2.5 | −0.8 to −2.5 |
| Magnification | 81000 | 22500 | 22500 | 81000 |
| Reconstruction | | | | |
| Software | cryoSparcV3.3 | cryoSparcV3.3 | cryoSparcV3.3 | cryoSparcV3.3 |
| Particles | 109,878 | 43,577 | 47,597 | 164,661 |
| Symmetry | C1 | C1 | C3 | C1 |
| Box size (pix) | 300 | 256 | 256 | 400 |
| Resolution (Å) ($FSC_{0.143}$) | 4.70 | 6.58 | 5.82 | 5.4 |
| Refinement | | | | |
| Software | Phenix 1.19 | | | |
| Protein residues | 12269 | | | |
| Chimera CC | 0.82 | | | |
| EMRinger Score | 0.97 | | | |
| R.m.s. deviations | | | | |
| Bond lengths (Å) | 0.007 | | | |
| Bond angles (°) | 0.776 | | | |
| Validation | | | | |
| Molprobity score | 2.19 | | | |
| Clash score | 6.89 | | | |
| Favored rotamers (%) | 99.1 | | | |
| Ramachandran | | | | |
| Favored regions (%) | 91.1 | | | |
| Disallowed regions (%) | 0.5 | | | |

analysis has also been carried out for a detergent-solubilized LASV trimer, in which the signal peptide forms an integral part of the transmembrane spanning region and the native cleavage site between GP1 and GP2 enables apex binding of matriglycan[15]. Recently, the I53-50A nanoparticle system[41] has been used to make nanoparticles of the LASV GPC trimer, which elicited neutralizing responses in rabbits, though not in guinea pigs[18]. Could a modified immunogen elicit higher titer neutralizing responses – and which sites on the GPC trimer might be more amendable to eliciting neutralizing responses?

We employed structure-based design to engineer an inter-protomer disulfide bond and to append a foldon trimerization domain to obtain a soluble ligand-free LASV GPC trimer stabilized in its prefusion state. Placement of the LASV GPC trimers in nanoparticles and immunization of guinea pigs elicited LASV-neutralizing responses. While the response showed elicitation of high titers, it did require a number of injections that would likely be impractical for a real-world vaccination schedule. Nonetheless, the neutralizing response observed here provides a starting point around which to optimize the vaccination strategy. Panning of the stabilized LASV GPC trimer against phage libraries identified a single domain antibody, D5, which bound the stabilized trimer and neutralized LASV in a bivalent IgG2a format, recognizing a glycan-free hole at the trimer apex. The apex-situated site-of-vulnerability appeared to be specific to GPCs with partial clea-vage between GP1 and GP2 subunit, that is, to be specific to a cleavage-intermediate form of the Lassa GPC trimer.

Neutralization through one-antibody-per-trimer stoichiometry with binding at the trimer apex –like D5– has been observed with other antibodies targeting other viral envelope trimers. These include PG9[42], PGT145[43], PGDM1400[44] and CAP256-VRC25.26[45,46], which neutralize HIV-1 potently, as well as antibodies PIA174[47] and PI3-E12[48], which neutralizes PIV3 potently. The recognition of such apex-binding anti-bodies offers a three-fold advantage in stoichiometry, and we note that in the case of D5, its neutralization was higher versus its binding affi-nity, when compared to other nanobodies that bound our GPC trimer.

The cryo-EM structure of D5 bound to the LASV GPC highlights an asymmetric cleavage intermediate that exposes a site-of-vulnerability at the trimer apex. We speculate that this conformation is available in an intermediate state resulting from a partially incomplete SP1-cleavage process. Although there is no direct evidence for a cleavage intermediate on a native virus, it is clear that this conformation and/or state must be present on GPC trimers on the virus surface, since D5 is able to neutralize. Alternatively, native trimer instability may play a role in exposing this vulnerable conformation and/or state on the virion surface. "Breathing" or flexibility around the apex cavity of a fully cleaved GPC trimer could account for D5 neutralization. We also note that although the native cleavage sequence near the D5 binding site is not present in our construct, D5 is able to neutralize and that binding of the nanobody would sterically occlude the binding of matriglycan. The asymmetry of the trimer may prevent elicitation of quaternary antibodies like 8.9F[17]; however, the resulting exposure of

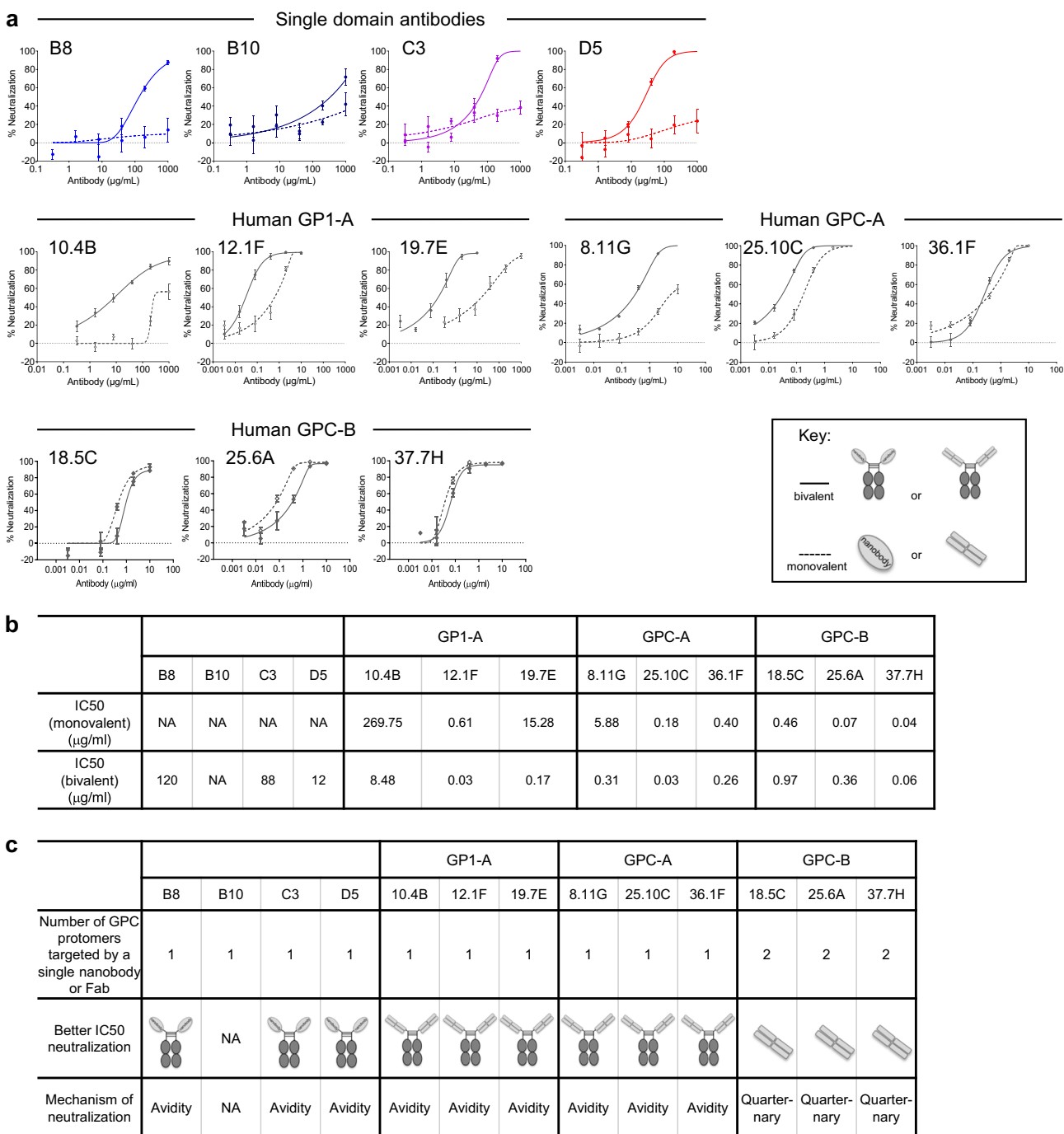

**Fig. 4 | Neutralization of Lassa virus by single domain antibodies and most human neutralizing antibodies utilizes avidity. a** Neutralization of pseudotyped Josiah strain of Lassa virus by single domain antibodies and human Lassa neutralizing antibodies in both monovalent (dotted line) and bivalent (solid line) formats. Neutralization was determined by interpolation after fitting data globally to a five-parameter dose-response curve. The IC50 (dot) and 95% CI (error bar) of a global fit of six serial dilutions. **b** Summary of the IC50 values of single domain antibodies and human Lassa neutralizing antibodies in monovalent and bivalent formats. **c** Summary of the proposed neutralization mechanisms of single domain antibodies and human antibodies for Lassa virus.

regions of the apex cavity may also elicit neutralizing antibodies that are more able to occlude the matriglycan-binding site.

The neutralization that our GPC-trimer nanoparticle elicited in guinea pigs was focused primarily on the Josiah and Pinneo lineages. While neutralizing titers were likely sufficient to protect from infection by viruses of these two lineages[18,24], it will nonetheless be critical to increase titer against other circulating LASV lineages.

In summary, we have stabilized a GPC trimer antigen in the absence of antibody, screened for neutralizing nanobodies,

determined an asymmetric trimer structure with a novel apex site-of-vulnerability, and demonstrated the elicitation of LASV-neutralizing responses in both guinea pigs and rhesus macaques, with peak titers averaging over 500 $ID_{50}$ in guinea pigs (Supplementary Fig. 3). While our trimer as well as the GPC-I53-50NP both deviate from the native trimer by altering the receptor-binding site in either sequence or conformation, both were able to elicit neutralizing responses (Supplementary Table 1). It remains to be determined if our cleavage-intermediate LASV trimer-nanoparticles can be modified to increase

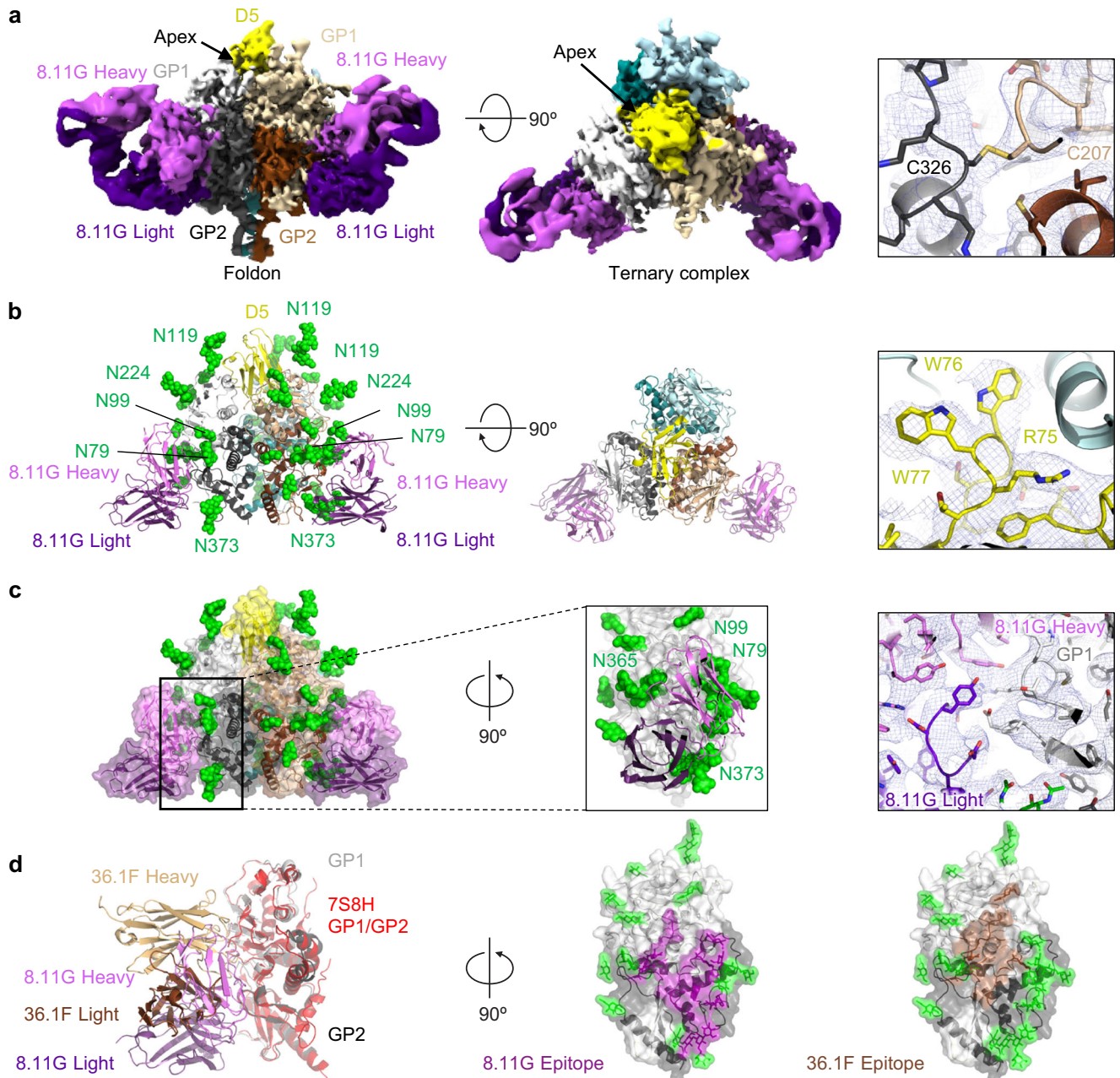

**Fig. 5 | Cryo-EM structure of D5 and 8.11 G with Lassa trimer reveals details of 8.11 G and D5 recognition. a** Cryo-EM density is shown for a complex of the stabilized GPC trimer bound to two Fabs of 8.11 G and a single D5. Density for the R207GC-L326C engineered disulfide is shown in the right panel. **b** The atomic model is shown in cartoon representation. Glycans for which density was observed are displayed as green spheres. A 90-degree view from the top is shown without glycans in the center. In the right panel is density around the D5 nanobody chain.

**c** The glycosylated epitope of 8.11 G is highlighted. The structure is displayed in cartoon format with a transparent surface. The inset box shows a view rotated 90° along the axis of the 8.11 G Fab binding with Fab contact loops shown as cartoon and the GPC protein and glycans shown as spheres. The right panel shows representative density of the antibody epitope. **d** The 8.11 G epitope partially overlaps with that of 36.1 F (PDB ID: 7S8H). A 5 A footprint of each antibody is shown.

their elicitation of heterologous neutralizing titers against LASV lineages II and III that make up a substantial portion of the currently observed cases of LASV[49,50].

## Methods

### Structure-based design of stabilized LASV GPC trimer

Designs of inter-protomer disulfide bond and foldon trimerization domain in LASV GPC were performed based on the LASV GPC-37.7H Fab complex structure (PDB ID: 5VK2[11]). In total, 164 design variants were made, including 68 with disulfide bonds, 43 with cavity-filling mutations, 22 with helix-breaking mutations, and 31 with trimerization

domain insertions with linkers of various lengths (Supplementary Fig. 1).

### Antigenic screening of LASV GPC stabilizing designs

Assessment of all 164 constructs was performed using a high-throughput 96-well microplate expression format followed by an ELISA-based antigenic evaluation[51]. Briefly, 100 μl/well of log-phase growing HEK 293 T cells was seeded into a flat bottom 96-well tissue culture plate (Corning) at a density of $2.5 \times 10^5$ cells/ml in an optimized expression medium (RealFect-Medium, ABI Scientific) and incubated at 37 °C, 5% $CO_2$ for 24 h. Prior to transfection, 40 μl/well

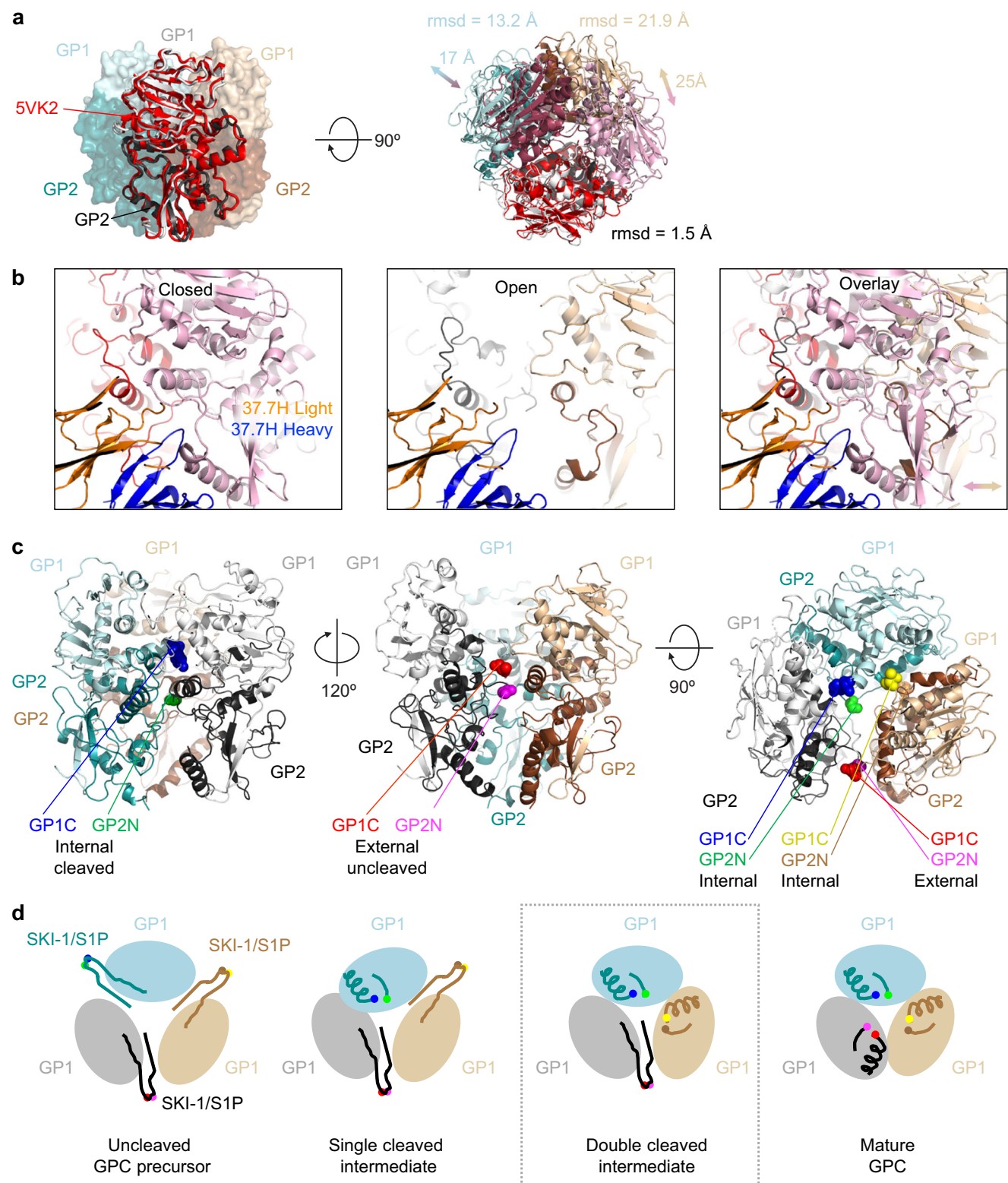

of spent medium was removed. For transient transfection, 0.25 ug/well of plasmid DNA in 10 μl of Opti-MEM medium (Invitrogen) in a round bottom 96-well plate was mixed with 0.75 μl TrueFect-Max transfection reagent (United BioSystems) in 10 μl of Opti-MEM medium. After incubation for 15 min at room temperature (RT), 20 μl/well of DNA-TrueFect-Max complex was mixed with growing cells in the 96-well tissue culture plate and incubated at 37 °C, 5% $CO_2$. After 12 h post transfection, the cells were fed with 30 μl/well

of enriched expression medium (CelBooster Cell Growth Enhancer Medium for Adherent Cell, ABI Scientific). The transfected cells were cultured at 37 °C, 5% $CO_2$ for five days. Supernatants with the expressed LASV GPC variants were harvested and tested by ELISA for binding to 37.7H antibody using $Ni^{2+}$-NTA microplates. HEK 293 T cells, Expi293F and FreeStyle 293-F cells used in this study were acquired commercially through Thermo Fisher Scientific.

**Fig. 6 | Analysis of asymmetric Lasa GPC trimer reveals a cleavage-intermediate site of vulnerability at the trimer apex. a** A single protomer of the GPC trimer shows a closely matching rmsd with that of the C3 symmetric GPC (PDB ID 5VK2, colored red, pink, and raspberry). The C1 symmetric GPC trimer observed here does not maintain the same quaternary assembly with the adjacent protomer oriented to accommodate the uncleaved protomer. After removing outlying residues the rmsd for one GP1/GP2 across 294 Ca atoms of the protomer is 1.5 Å, however the rmsd's for the other two protomers are 13.2 and 21.9 Å across 296 and 294 Ca, respectively. Specific apex regions showing distances from 17-25 Å. **b** Explanation for the loss of one, two, or three of the 37.7H binding sites on a trimer with D5 stabilizing a more open interface between protomers. The open gap that

accommodates uncleaved peptide is shown compared to the closed epitope interface. **c** The Lassa GPC is shown with focus on one of the two cleaved protomers (left) with the internal termini of GP1 (light cyan) and GP2 (teal) shown. Rotation by 120° shows the external location of the uncleaved peptide. **d** A schematic representation of the cleavage intermediates in the maturation of the GPC trimer is shown from a top view looking down the trimer axis. The SKI-1/S1P cleavage site must be cleaved on all three protomers to enable a tightly packed GPC trimer. The neutralizing nanobody D5 binds the first three populations (uncleaved, single-cleavage, and double-cleaved). Gray box highlights the intermediate corresponding to the structure shown above.

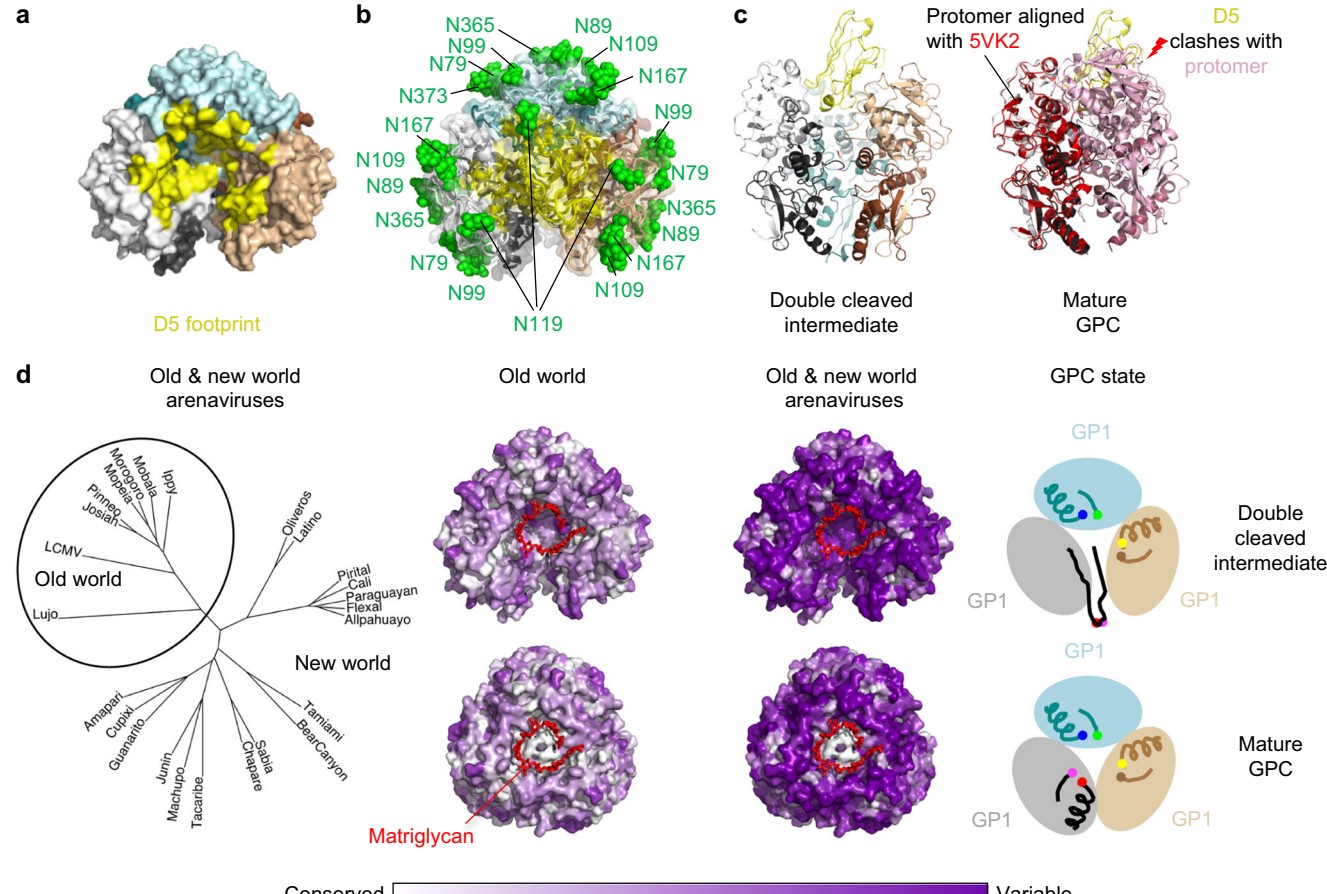

**Fig. 7 | Apex site of nanobody vulnerability overlaps with the binding site for matriglycan receptor. a** A surface representation of the cleavage intermediate trimer is shown. Residues within a 5 Å footprint of the D5 nanobody are highlighted in yellow. **b** A transparent surface representation is shown as in panel A with cartoon representation underneath. Glycans are displayed as green spheres. **c** D5 fits into pocket at apex of double-cleaved intermediate, but not into the mature GPC, in which one protomer orientation shifts the apex region by >10 Å (pink) and the

second protomer shifts at apex regions by >20 Å (raspberry), sterically blocking access to the the D5 binding site. **d** (left) A phylogenetic tree of select arenavirus GPC sequences is displayed to illustrate the overall sequence diversity and highlight the clustering of old world versus new world viruses. (right) Conservation of the GPC residues is shown for the old world viruses versus all arenaviruses with the double-cleavage intermediate on the top and the mature GPC on the bottom, the matriglycan (red) from PDB ID 7PVD is shown on all surfaces.

### Enzyme-linked immunosorbent assay ELISA
Briefly, Ni$^{2+}$-NTA microplates (Pierce) were incubated with 30 μl/well the supernatant of LASV GPC variants mixed with 70 μl/well of PBS at 4 °C overnight, followed by blocking with 200 μl/well of CelBooster Cell Growth Enhancer Medium for Adherent Cell for 1 h at RT. Between each subsequent step, plates were washed five times with PBS-T (PBS plus 0.05% Tween 20). After plate wash, 100 μl/well of 10 μg/ml GPC-specific antibody 37.7H was incubated at RT for 2 h. followed by incubation for 30 min at RT with 100 μl/well of horseradish peroxidase (HRP) conjugated goat anti-human IgG antibody (Jackson ImmunoResearch Laboratories Inc., PA), diluted at 1:10,000 (v/v) in CelBooster Cell Growth Enhancer Medium for Adherent Cell plus

0.02% Tween 20. Then, the reaction signal was developed with 100 μl/well of tetramethylbenzidine substrate (BioFX-TMB, SurModics) for 10 min at RT before the addition of 100 μl/well of 0.5 N sulfuric acid (Fisher Chemical) to stop the reaction. Plates were read at 450 nm wavelength (SpectraMax using SoftMax Pro, version 5, software; Molecular Devices, Sunnyvale, CA), and the optical densities (OD) were analyzed following subtraction of the nonspecific horseradish peroxidase background activity. All samples were measured in duplicate.

### LASV GPC trimer protein expression and purification
Disulfide and foldon-stabilized LASV GPC sequence was attached to a thrombin cleavage sequence, a hexahistidine tag, and a Strep-tag at its

C-terminal end. The stabilized LASV GPC protein was expressed by transient transfection in 293 F cells (Thermo Fisher) with Turbo293 transfection reagent (SPEED BioSystem) using the established protocol[52]. Briefly, one liter of 293 F cells at a density of $1.2 \times 10^6$ cells/ml were co-transfected with 700 µg/liter of the LASV GPC expression plasmid and 300 µg/liter of furin plasmid. Six days post transfection, the culture supernatant was harvested and protein was purified from the supernatants by Nickel- (Roche) and Strep-Tactin- (IBA lifesciences) affinity columns. The resulted protein was loaded on a Superdex 200 16/600 size exclusion column (GE Healthcare) to be further polished for use in subsequent assays. SDS-PAGE gels were run to assess protein purity and disulfide formation. Gels in figures were cropped to show the area of interest, full uncropped gel image scans are present in the Source Data file.

## Production of nanoparticle immunogens

We used the SpyTag/SpyCatcher system to create the Lassa GPC encapsulin nanoparticles. The gene of encapsulin with spyCatcher (EN-spyC) was codon optimized and cloned into a pET11a vector (Novagen), and two cysteines were introduced at positions 53 and 94 to enable disulfide stabilization of the assembled nanoparticle. The plasmid was transformed into Shuffle T7(New England Biolab) cells. The cells were induced with 0.2 mM IPTG overnight at 18 °C at an OD of 0.6. Harvested bacterial cells were lysed by sonication, and supernatant was heated at 56 °C for 15 min. Supernatant was clarified by centrifuge, and saturated (NH4)2SO4 solution was added to final concentration of 20% saturation. Precipitates were harvested and re-suspended in PBS. After removal of endotoxin, encapsulin nanoparticle were purified by size exclusion chromatography (Superdex 200 Increase 10/300 GL) in PBS. To prepare Lassa GPC trimer nanoparticles, EN-spyC and Lassa GPC trimer-spyT were mixed at a 1:1.2 molar ratio and incubated for ~2 h. Nanoparticle were purified by size exclusion column on Superdex200 Increase 10/300 GL in PBS buffer, and assess by negative strain EM for their incorporation of GPC trimers.

## Production of human LASV antibodies

Immunoglobulin heavy chain or light chain sequences were constructed by gene synthesis and then cloned into human IgG1, lambda, or kappa expression plasmids as previously described[53]. Heavy and light chain expression plasmid DNA was transfected into Expi293F cells (Thermo Fisher) in 1:1 (v/v) ratio using Turbo293 transfection reagent[54]. Monoclonal antibodies from the culture supernatants were purified using recombinant Protein-A Sepharose (GE Healthcare) as per the manufacturer's instructions.

## Antibody Fab preparation

The purified human IgG proteins were cleaved by LysC enzyme (1:4000 w/w) (Roche) at 37 °C overnight to yield Fabs. On the next day, the enzymatic digestion reaction was terminated by addition of protease inhibitor (Roche). The cleavage mixture was then passed through a Protein-A column to separate the Fc fragments from the Fab. The Fab collected in the flow-through was loaded onto a Superdex 200 16/60 column for further purification.

## LASV GPC antigenic characterization

An Octet Red384 instrument (fortéBio) was used to measure the binding kinetics between the stabilized LASV GPC trimers and human LASV neutralizing antibodies or nanobodies. Assays were performed at 30 °C in tilted black 384-well plates (Geiger Bio-One). Ni-NTA sensor tips (fortéBio) were used to capture the histidine-tagged stabilized LASV GPC trimer for 300 s. Then, the biosensor tips were equilibrated for 60 s in PBS before measurement of association with antigen-binding fragments (Fabs) in solution (6.25 nM to 400 nM) for 180 s. Subsequently, Fabs were allowed to dissociate for 300 s. Parallel correction to subtract systematic baseline drift was carried out by subtraction of the measurements recorded for a loaded sensor dipped in PBS. Data analysis and curve fitting were carried out using the Octet Data Analysis Software 9.0 (fortéBio). Experimental data were fitted with the binding equations describing a 1:1 interaction. Global analysis of the data sets assuming reversible binding (full dissociation) were carried out using nonlinear least-squares fitting allowing a single set of binding parameters to be obtained simultaneously for all of the concentrations used in each experiment.

## Negative-stain electron microscopy

The protein was diluted with a buffer containing 10 mM HEPES, pH 7.0, and 150 mM NaCl to a concentration of 0.02 mg/ml and adsorbed to a freshly glow-discharged carbon-coated copper grid. The grid was washed with the same buffer, and the adsorbed protein molecules were negatively stained with 0.7% uranyl formate. Micrographs were collected at a nominal magnification of 100,000 using SerialEM[55] on an FEI T20 electron microscope equipped with a 2k x 2k Eagle camera and operated at 200 kV. The pixel size was 0.22 nm. Particles were picked automatically using in-house written software (YT, unpublished) and extracted into 100 × 100-pixel boxes. Reference-free 2D classifications were performed using Relion[56].

## Physical stability of the designed LASV GPC trimer

To assess the physical stability of the designed LASV GPC trimer under various stress conditions, we treated the proteins with a variety of pharmaceutically relevant stresses such as extreme pH, high temperature, low and high osmolarity, and repeated freeze/thaw cycles while at a concentration of 50 µg/ml. The physical stability of treated LASV GPC trimer was evaluated by the preservation of binding to the GPC-specific antibody 37.7H. Temperature treatments were carried out by incubating the stabilized LASV GPC protein solutions at 50 °C, 70 °C and 90 °C for 60 min in a PCR cycler with heated lid. In pH treatments, the stabilized LASV GPC protein solution was adjusted to pH 3.5 and pH 10.0 with appropriate buffers for incubation at room temperature for 60 min and subsequently neutralized to pH 7.5. In osmolarity treatments, the stabilized LASV GPC protein solutions originally containing 150 mM NaCl were either diluted with 2.5 mM Tris buffer (pH 7.5) to an osmolarity of 10 mM NaCl or adjusted with 4.5 M $MgCl_2$ to a final concentration of 3.0 M $MgCl_2$. Protein solutions were incubated for 60 min at room temperature and then returned to 150 mM salt by adding 5.0 M NaCl or dilution with 2.5 mM Tris buffer, respectively, and concentrated to 50 µg/ml. The freeze/thaw treatment was carried out by repeatedly freezing the stabilized LASV GPC protein solutions in liquid nitrogen and thawing at 37 °C ten times. The degree of physical stability is reported as the ratio of steady state 37.7H antibody-binding level before and after stress treatment.

## Animal protocols and immunization

All animal experiments were reviewed and approved by the Animal Care and Use Committee of the Vaccine Research Center at the NIAID (NIH). Animals were housed and cared for in accordance with local, state, federal, and institute policies in an American Association for Accreditation of Laboratory Animal Care-accredited facility at the Vaccine Research Center.

Female Hartley guinea pigs with body weights of >300 g were purchased from Charles River Laboratories (Wilmington, MA). For each immunization, 25 ug immunogen resuspended with Adjuplex (Advanced Bio Adjuvants LLC, Omaha, NE) in PBS, and was injected via a needle syringe to the caudle thigh of the two hind legs. Blood was collected for serological analyses.

Five to twelve year old, male and female rhesus macaques, weighing between 5-10 kgs were used in this study. For each immunization, 100 µg immunogen mixed with 20% of Adjuplex in final 1 mL PBS and was injected via a needle syringe combination into the caudal thighs of the two hind legs. Blood was collected 2 weeks post immunization for serological analyses.

## Isolation and B cell sorting of guinea pig peripheral blood mononuclear cells (PBMCs)

3-4 ml of Guinea pig whole blood was centrifuged at 900 g for 10 min at room temperature. After removing the plasma on the top, the cell pellet was resuspended in 8 ml of PBS/2 mM EDTA and carefully layered on top of a 3 ml Lympholyte-Mammal (Cedarlane, #CL5115) cushion in a 15 ml conical tube. Following a 20-min centrifugation at 1000 g in a swing bucket, collect the white PBMCs at the buffer/Lympholyte interface, dilute in 10 ml of PBS, and spin down the cells at 900 g for 10 min. Resuspend the cells in 3 ml of 90% heat inactivated Fetal Bovine Serum/10% DMSO and freeze them down in 1 ml aliquots using a Mr. Frosty Cryo freezing container at −80 °C for overnight. The PBMCs can then be transferred to liquid nitrogen tank for long term storage. For antigen-specific B cell sorting, PBMCs were thawed in a 37 °C water bath, washed once with PBS and stained with the LIVE/DEAD™ Fixable Violet Stain (Thermofisher #L34955) following manufacturer's instructions. After PBS wash, the cells (2-5 million) were further stained in 100 µl of a staining mix containing 0.6 µl mouse-anti-GP T Lymphocytes_FITC (BioRad #MCA751F), 0.5 µl goat anti-GP IgG_Ax594 (RRID: AB_2337442), 0.5 µl MERS-S2P-foldon_PECy7 (negative probe for foldon), 2 µl LASV-GPC-foldon_APC and 2 µl LASV-GPC-foldon_PE in PBS/1% FBS on ice in dark for 30 min. The stained cells were washed twice before being resuspended in 0.5 ml PBS/1%FBS and filtered through 40 µm mesh. In order to reach the lower limit of cell counts for 10x Genomics – 4000 cells, we first bulk sorted 4000 T cells and then sorted 150 - 500 LASV-GPC⁺/MERS-S2P-foldon⁻/IgG⁺ B cells from each guinea pig sample.

## Cloning of guinea pig anti-LASV GPC antibodies by 10x Genomics

The combined bulk sorted T and LASV-GPC specific B cells are processed for 10x Genomics Chromium Single Cell BCR sequencing (following manufacturer's instructions) using the 5′ RACE primer and the following guinea pig 3′ primers for amplifying IgG heavy and light chains.

| |
|---|
| GP_JH1/2/5_R: CTGGCTGAGGAGACGGTGACCAG |
| GP_JH3_R: CTGGCTGACGTGACGGTGACTGA |
| GP_JH4_R: CTGGCTGAGGAGACGGTGACCGA |
| GP_JH6_R: CTGGCTGAGGAGATAGTGACCAG |
| GP_IgK_R: GAAGAGGGAGATAGTTGGCTTCTGCACACT |
| GP_IgL1_R: GAGGAGGGYGGGAACAGGCTGACTGTGG |
| GP_IgL2_R: GAGGAGGGTGGGAATAGGCTGACTGTGG |

From 4 immunized guinea pigs, 435 pairs of IgG heavy and light chain sequences were obtained, gene synthesized and cloned into our guinea pig IgG mammalian expression vectors (VRC6705 for Gamma, VRC6479 for Kappa and VRC6480 for Lambda). These IgGs were each transiently expressed in 200 µl of 293 T cells in 96 well plates and the day 6 supernatants were screened by ELISA against LASV GPC (Josiah) and a negative control foldon-containing protein RSV-F.

## Phage display panning of nanobody libraries

Shark V_NAR and camel V_HH nanobody phage display libraries were constructed in Dr. Mitchell Ho's laboratory at the National Cancer Institute (Bethesda, Maryland) according to the protocol described previously[29]. The phage panning protocol has been described previously[57,58]. Briefly, an immunotube (Nunc/Thermo Fisher Scientific, Rochester, NY) was coated with 0.5 ml of 10 µg/ml LASV GPC trimer in PBS at 4 °C overnight. After decanting the coating buffer, the immunotube was treated with 0.5 ml blocking buffer (10% milk in PBS) at room temperature for 1 h. Then, a fixed amount of input phage from the shark or camel libraries was added to the immunotube for binding to the LASV GPC trimer at room temperature for 2 h with gentle

shaking. The immunotube was washed with PBS containing 0.05% Tween-20 to remove unbound phages. Subsequently, the bound phages were eluted with 100 mM triethylamine. At each round of panning, output phage enrichments were assessed and monitored by polyclonal phage ELISA. Single colonies were picked at the final round of panning for DNA sequencing. The binding ability of the identified nanobodies from phage display toward the stabilized LASV GPC trimer was further evaluated by ELISA, with bovine serum albumin (BSA) serving a negative control.

## Polyclonal phage ELISA

A 96-well plate (Corning) was coated with 50 µl/well of 5 µg/ml stabilized LASV GPC trimer in PBS buffer at 4 °C overnight. After blocking with 3% milk in 100% superblock buffer (Thermo Scientific) at room temperature for 2 h, 50 µl phage were added to the plate and incubated at room temperature for another hour. Binding of phage to the stabilized LASV GPC was detected by HRP-conjugated anti-M13 antibody (GE Healthcare). The cut-off value for positive binder was set as 3 × higher signal of antigen binding compared to background signal.

## Expression of nanobodies

The pComb3x phagemids containing the V_NAR or the V_HH binders were transformed into HB2151 E. coli cells. The formed colonies were pooled for culture in 2 L 2YT media containing 2% glucose, 100 µg/ml ampicillin at 37 °C until the OD600 reaches 0.8–1. Culture media was then replaced with 2YT media containing 1 mM IPTG (Sigma), 100 µg/ml ampicillin, and shook at 30 °C overnight for soluble protein production. Bacteria pellet was spun down and lysed with polymyxin B (Sigma Aldrich) for 1 h at 37 °C to release the soluble protein. The supernatant was harvested after lysis and purified using HisTrap column (GE Healthcare) using AKTA.

## Protein ELISA

Protein ELISA was used to evaluate the binding ability of the selected nanobody binders toward the stabilized LASV GPC trimer. Briefly, a 96-well plate was coated with either the stabilized LASV GPC trimer or BSA at 5 µg/ml in PBS, 50 µl/well, at 4 °C overnight. After blocking with 100% superblock buffer, the nanobodies were diluted into 1 µg/ml using 10% PBST in 100% superblock and then added to the plate for incubation at room temperature for 1 h. Binding signal was detected by HRP-conjugated anti-Flag antibody (Sigma). For the competition ELISA, neutralizing antibodies D5, 19.7E, 8.11 G, and 37.7H were used to block the binding epitopes. An anti-Foldon antibody, MF4 were also used as a control.

## Cross-competition assay

The histidine- and Strep-tagged stabilized LASV GPC trimer protein (30 µg/ml) was captured by a mouse anti-streptavidin antibody, which was immobilized by the anti-mouse Fc sensor tips to a final mean signal level of 1.0-1.5 nm. The trimer-coated tips were then dipped into either PBS or the pre-determined saturating concentrations of Fabs (1000 nM) or nanobodies (500 nM) (first ligand) in PBS for 300 s. After loading, the sensor tips were incubated in PBS briefly for 60 s to remove unbound ligands for baseline adjustment. Subsequently, the sensor tips were dipped into wells containing a fixed concentration of competing ligands (second ligand, 1000 nM Fabs or 500 nM nanobodies) for another 300 s, followed by 300 s of dissociation in PBS. Raw data was processed using Octet Data Analysis Software 9.0. Percent of residual binding was calculated as follows: (response signal from the second ligand in presence of first ligand / response signal from the second ligand in absence of first ligand) x 100.

## Generation of nanobody-IgG2a proteins

To express nanobodies in a bivalent IgG format, the gene encoding nanobody variable region was cloned into the mammalian protein

expression vector pVRC8400 in front of DNA sequences encoding an Ala-Ala-Ala linker, the llama IgG2a hinge sequence (EPKIPQPQPKPQPQPQPQPKPQPKPEPECTCPKCP) and the human IgG1 Fc domain. The nanobody IgG2a proteins were expressed by transient transfection in 293 F cells (Thermo Fisher) with Turbo293 transfection reagent (SPEED BioSystem) using the protocol described above and purified by protein A affinity column.

## Production of pseudovirus

Recombinant Indiana VSV (rVSV) expressing LASV GPC were generated as previously described[59,60]. HEK293T cells were grown to 80% confluency before transfection with plasmids expressing LASV Josiah GPC using FuGENE 6 (Promega). Cells were cultured at 37 °C with 5% $CO_2$ overnight. The next day, medium was removed and VSV-G pseudotyped ΔG-luciferase (G*ΔG-luciferase, Kerafast) was used to infect the cells in DMEM at a MOI of 3 for 1 h before washing the cells with 1 × DPBS three times. DMEM supplemented with 2% fetal bovine serum and 100 I.U./mL penicillin and 100 μg/mL streptomycin was added to the infected cells and they were cultured overnight as described above. On the following day, the supernatant was harvested and clarified by centrifugation at 300 g for 10 m before aliquots and storage at −80 °C.

## Pseudovirus-based neutralization assay

Neutralization assays were performed by incubating pseudoviruses with serial dilutions of antibodies and measured by the reduction in luciferase gene expression. In brief, Vero E6 cells (ATCC) were seeded in a 96-well plate at a concentration of $2 \times 10^4$ cells/well the day before. Pseudoviruses were incubated with serial dilutions of antibodies (six dilutions in a 5-fold stepwise manner) in triplicate at 37 °C for 30 min. Then, the mixture was added to cultured cells for infection and incubated for an additional 24 h. The luminescence was measured by Britelite plus Reporter Gene Assay System (PerkinElmer). Neutralization titers (ID50: 50% inhibitory dose or IC50: 50% inhibitory concentration) were defined as the serologic sample dilution or antibody concentration at which the relative light units (RLUs) were reduced by 50% compared with the virus control wells (virus + cells) after subtraction of the background RLUs in the control groups with cells only. The IC50 or ID50 values were calculated with non-linear regression using GraphPad Prism 8 (GraphPad Software, Inc.).

## Cryo-EM data collection and processing

The prefusion-stabilized Lassa GPC trimer was deposited on a grid unliganded or incubated with 2-fold molar excess per protomer of nanobody D5 and antibody Fab fragments of 8.11 G. A volume of 2.3 μl of the complex at 1 mg/ml concentration was deposited on a C-flat grid (protochip.com) and the grid was vitrified using an FEI Vitrobot Mark IV with a wait time of 30 s, blot time of 3 s and blot force of 1. Data collection was performed on a Titan Krios electron microscope with Leginon[61] with a Gatan K2 Summit direct detection device for the unliganded trimer and a Gatan K3 for the liganded sample. Exposures were collected in movie mode for a 10 s with the total dose of 56.52 e$^-$/Å$^2$ fractionated over 50 raw frames for the unliganded and for a 2 s with the total dose of 51.15 e$^-$/Å$^2$ fractionated over 50 raw frames for the liganded. Images were pre-processed using Appion[62,63]; individual frames were aligned and dose-weighted using MotionCor2[64]. CTFFind4[65,66] was used to estimate the CTF and DoG Picker[62,63] was used for particle picking. RELION[56] was used for particle extraction. CryoSPARC 3.3[67] was used for 2D classifications, ab initio 3D reconstruction, homogeneous refinement, and nonuniform 3D refinement. The unliganded trimer showed several C1 classes and we refined the map of the class that had the conformation of those containing a single uncleaved protomer interface. Likewise, the antibody bound datasets showed a number of classes with one, two, or three fabs bound and one or two uncleaved interfaces, the best-resolved class contained a single uncleaved interface, two fabs, and a single D5 nanobody.

Coordinates from PDB ID 5VK2 was used for initial fit to the reconstructed map. This was followed by simulated annealing and real space refinement in Phenix[68] and then iteratively processed with manual fitting of the coordinates in Coot[69]. Geometry and map fitting were evaluated throughout the process using Molprobity[70] and EMRinger[71]. PyMOL (www.pymol.org) and ChimeraX[72] were used to generate figures.

## Data availability

The cryo-EM maps generated in this study have been deposited to the Electron Microscopy Data Bank with accession codes EMD-26859 for ligand-free Lassa GPC trimer with C3 symmetry, EMD-26740 for ligand-free Lassa GPC trimer with C1 symmetry, EMD-41302 for Lassa GPC trimer in complex with Fab GP23, and EMD-41048 for Lassa GPC trimer in complex with Fab 8.11 G and nanobody D5. The coordinates for the cryo-EM structure of the Lassa GPC trimer in complex with Fab 8.11 G and nanobody D5 have been deposited in the Protein Data Bank with accession code 8T5C. Source data for Figs. 1 and 2 are available in the Source Data.xlsx file. Source data are provided with this paper.

## Code availability

The in-house written software for automatic particle picking from negative-stain EM images is available upon request to Yaroslav Tsybovsky.

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

## Acknowledgements

We thank M. Sastry for assistance with control antibodies, J. Stuckey for assistance with figures, D. Ambrozak for assistance with B cell sorting, R. Woodward and the Translation Research Program of the Vaccine Research Center for assistance with animal studies, and members of the Structural Biology Section and Structural Bioinformatics Core, Vaccine Research Center, for discussions and comments on the manuscript. Support for this work was provided by the Vaccine Research Center, an intramural Division of the National Institute of Allergy and Infectious Diseases (NIAID), National Institutes of Health (NIH), and by the Intramural Research Program of the Center for Cancer Research, National Cancer Institute, NIH. This work was also supported in part with federal funds from the Frederick National Laboratory for Cancer Research, NIH, under contract HHSN261200800001E. Some of this work was performed at the Simons Electron Microscopy Center and National Resource for Automated Molecular Microscopy located at the New York Structural Biology Center, supported by grants from the Simons Foundation (SF349247), NYSTAR, and the National Institute of General Medical Sciences, NIH (GM103310).

## Author contributions

J.G. and C.S-F.C. co-headed the design of a soluble GPC trimer, with L.O., J.C.B., B.Z., Y.Y. and T.Z. contributing to the immunogen design. J.G. determined and analyzed cryo-EM structures. C.S-F.C, Z.D., L.O., I.T.T., R.V., D.W., C-Y.Z., S.J.C., Z.W., T.L., B.Z., expressed and purified immunogens and antibodies. T.S., Y.T. performed negative staining analysis. C.C., J.P.T. and L.O. co-headed the animal studies. M.W. and P.W. performed and analyzed neutralization. E.K.S., Y.Y., H.D., and A. B. performed ELISA assays. X.C. and S.C. performed B-cell sorting and antibody identification. A.R.H. and C.A.S. performed antibody sequences analysis. Z.D. and Y.S. performed nanobodies panning with shark VNAR and camel VHH libraries. T.B. and C-H. S. performed sequence analyses. J.G., C-S-F.C, S.W., and P.D.K. wrote the original manuscript draft. All authors reviewed and edited the manuscript. D.C.D., J.R.M., D.D.H, M.H., and P.D.K. supervised the study and reviewed all data.

## Competing interests

NIH has submitted a patent application for GPC trimer and GPC trimer nanoparticle on which J.G., C-S-F.C., L.O., M.W., C.C., J.R.M., D.D.H, and P.D.K. are co-inventors. NIH has also submitted a patent application for neutralizing nanobodies on which J.G., C-S-F.C., Z.D., Y.S., M.H. and P.D.K. are co-inventors. The other authors declare no competing interests.

## Additional information

[1]Vaccine Research Center, National Institutes of Health, Bethesda, MD 20892, USA. [2]NCI Antibody Engineering Program, Center for Cancer Research, National Cancer Institute, National Institutes of Health, Bethesda, MD 20892, USA. [3]Aaron Diamond AIDS Research Center, Columbia University Vagelos College of Physicians and Surgeons, New York, NY 10032, USA. [4]Electron Microscopy Laboratory, Cancer Research Technology Program, Leidos Biomedical Research, Inc., Frederick National Laboratory for Cancer Research, Frederick, MD 21702, USA. [5]These authors contributed equally: Crystal Sao-Fong Cheung, Zhijian Duan, Li Ou, Maple Wang. ✉e-mail: dh2994@cumc.columbia.edu; homi@mail.nih.gov; pdkwong@nih.gov

