## [Peer Review File · Nature Communications]

Cleavage-Intermediate Lassa Virus Trimer Elicits Neutralizing Responses, Identifies Neutralizing Nanobodies, and Reveals an Apex-Situated Site-of-VulnerabilityREVIEWER COMMENTS

Reviewer #1 (Remarks to the Author):

Gorman and colleagues have built a LASV GPC trimer incorporating a novel inter-protomer disulfide bond. The majority of the stabilized trimer exists as a band of the size for a protomer, indicating that 1 and GP2 subunits were inefficiently cleaved. The protein is expressed with a final yield of approximately 0.5 mg/L suggesting that it could be used as a vaccine candidate.

The inter-protomer disulfide stabilized LASV GPC trimer is immunogenic and was able to elicit a neutralizing antibody and a neutralizing nanobody. The binding of the nanobody reveal a novel epitope (albeit one that overlaps with 8.9F} at the apex of partly GP1/2 cleaved trimers.

This is very solid well-presented work. There are limitations to the results that should have been discussed more fully.

While the construct did elicit neutralizing antibodies it took more than six injections. Such a vaccine requiring 5 boosters would not be practical in West Africa.

The nanobody D5 reveals a novel epitope, and this is of interest. However, the potency of the nanobody appears to be low even after making it a bispecific likely precluding it from being applied as an immunotherapeutic.

Minor point

Line 92 Sanders needs a reference – should be Brouwer and colleagues. –Also, this information is redundant with line 86 – please edit.

Reviewer #2 (Remarks to the Author):

The manuscript by Gorman et al. demonstrates that a prefusion-stabilized LASV GPC trimer was an improved antigen to elicit neutralizing antibodies, like GP23. A single domain antibody, D5, was found to recognize a glycan-free hole at the trimer apex and an asymmetrical cleavage intermediate of the GPC trimer is required to expose the “site-of-vulnerability”. In general, the manuscript has been prepared with high quality. I do have several comments that may improve the presentation.

Major points:

Cryo-EM studies of D5-GPC complexes were carried out by using stabilized GPC trimers. Unlike this engineered version, native GPC trimers may be able to interact with D4 in a more flexible manner and incomplete cleavage between GP1 and GP2 could be unnecessary. Unless there is more direct evidence, it would be better to modify the text in regarding of asymmetrical cleavage of GPC on the viral surface.

Minor points:

- 1) In Fig. 1f, all error bars look the same. Are they correct?
- 3) In Fig. 2a, the labels for protein bands on SDS-PAGE are misplaced.
- 4) in Fig. 3d, the colors for the 18.5C heavy and light chain are indistinguishable.
- 5) in Fig. 5, the GPC part shows high resolution, while the electron density for antibodies is relatively unclear. Is the resolution high enough for model building of antibodies?

Reviewer #3 (Remarks to the Author):

In the manuscript entitled "Cleavage-Intermediate Lassa Virus Trimer Elicits Neutralizing Responses, Identifies Neutralizing Nanobodies, and Reveals an Apex-Situated Site-of-Vulnerability", the authors engineered the glycoprotein complex (GPC) of Lassa virus based on reported structures and profound antigenic screening. The authors then characterized the engineered GPC trimer with known antibodies and demonstrated that the GPC construct in the nanoparticle format could elicit neutralizing antibodies in guinea pigs, and have successfully identified the first guinea pig-derived anti-LASV neutralizing mAb (GP23). Further, one potent neutralizing nanobody (D5) was sorted out by using the GPC construct. The aim of the project, which is the rational design of a suitable GPC construct for both immunization and neutralizing antibody discovery, is of great interest in the field, because the neutralizing antibody response is protective but extremely rare for Lassa. However, there are a couple of conflicts between the structural and biochemical results of the manuscript, as well as with previous studies, that need to be solved.

Major comments:

1. Page 4: in the paragraph describing the history of LASV GPC antigen design and production, one well-designed LASV GPC trimer, named GPCysRRLL (PMID: 36288283), was missed. This GPC antigen is a stabilized prefusion trimer with the native RRLC cleavage motif for the LASV receptor (matriglycan) recognition. Importantly, it is the only reported LASV GPC construct that is able to be recognized by anti-LASV human neutralizing antibodies from all four discovered human neutralizing epitope groups (GP1-A, GPC-A, GPC-B, and GPC-C), including 8.9F, an important preclinical therapeutic mAb which binds to the apex of GPC and directly blocks GPC/matriglycan receptor binding. Since one of the main features of this manuscript is GPC immunogen design for neutralizing antibody elicitation, it is recommended to include and discuss the GPCysRRLL construct in the text and table.S1.

2. Line 171: "...but gradually rose so that by the 5th and 6th nanoparticle immunization, they averaged about 500 IC50." The neutralizing titers of the immunized guinea pig sera are the crucial results of this manuscript. The measurement unit of the neutralizing titer is shown as "IC50", however, the authors didn't identify the "IC50" here. Is it from a specific monoclonal antibody or a serum sample? In addition, since there was one animal showed a much higher titer than others, it is recommended to use a median titer than an average titer. Plus, a typo in Fig. 2e: "ID50".

3. Line 204: "Binding competition revealed that nanobodies C3 and D5 competed for binding to GPC with known human LASV neutralizing antibodies from the GPC-B group". The epitope of the nanobody D5 binds to the apex of GPC trimer (Fig.5), which is far away from the epitope of GPC-B mAbs. Could the authors explain why D5 and GPC-B mAbs compete with each other?

4. Fig.5d: The structure models show the epitopes of 18.5C and 8.11G are overlapped, however, no binding competition is shown in Fig.S6C. In fact, 18.5C (GPC-B) and 8.11G (GPC-A) belong to two distinct epitope groups (PMID: 27161536).

5. About the neutralizing mechanism of nanobody D5: the authors found D5 only recognized the cleavage-intermediate GPC trimer but not the mature, fully cleaved trimer, and suggested it neutralized LASV by blocking receptor matriglycan. However, based on the previous structural study (PMID: 35173332), the binding of GPC/matriglycan required a mature GPC trimer with fully-cleaved RRLC motifs of all three protomers. Therefore, if D5 cannot recognize mature GPC, it wouldn't block the matriglycan binding for neutralization. On the other hand, even if the matriglycan is able to bind to the cleavage-intermediate GPC for virion cell entry, the uncleaved GP1/GP2 protomer(s) in the cleavage-intermediate GPC is not able to mediate virus-endosome membrane fusion. The author may dig deep to solve the issue on D5 nanobody. In addition, the "site-of-vulnerability" on the apex of the trimer has been discovered in the previous study (PMID: 36288283). It is not a new site.

6. For further evaluation, it is required to show the formal, complete PDB validation reports of the EM-maps and atomic models along with the manuscript.

Minor comments:

1. Page 5, line 89: "The highly glycosylated GPC induces weak and inconsistent immune responses in both natural infection and vaccination". Indeed, both T and B cell responses are potent after infection or immunization, however, just neutralizing antibody response is extremely weak and inconsistent.

2. Could the authors describe how many copies of GPC trimer are displayed on each encapsulating nanoparticle?

3. Fig. 3d: Could the authors show the side view of the EM-map?

4. Fig. S3a and b: What does the MF4 mean here?

5. Fig. 7d: Please display or write which colors represent the "Conserved" and "Variable" regions.

6. Typos:

Fig.2e and 3a: "ID50 neutralization"

Table S1: "G326C", should it be L326C?

Point-by-Point Response to Reviewers

Reviewer #1 (Remarks to the Author):

Gorman and colleagues have built a LASV GPC trimer incorporating a novel inter-protomer disulfide bond. The majority of the stabilized trimer exists as a band of the size for a protomer, indicating that 1 and GP2 subunits were inefficiently cleaved. The protein is expressed with a final yield of approximately 0.5 mg/L suggesting that it could be used as a vaccine candidate.

The inter-protomer disulfide stabilized LASV GPC trimer is immunogenic and was able to elicit a neutralizing antibody and a neutralizing nanobody. The binding of the nanobody reveal a novel epitope (albeit one that overlaps with 8.9F) at the apex of partly GP1/2 cleaved trimers.

This is very solid well-presented work. There are limitations to the results that should have been discussed more fully.

Response:

We appreciate this reviewer's summary of our study and the positive overall assessment.

While the construct did elicit neutralizing antibodies it took more than six injections. Such a vaccine requiring 5 boosters would not be practical in West Africa.

Response:

We agree with this reviewer that a vaccine with so many injections to elicit neutralizing responses is not optimal. We are only aware of one other publication eliciting any neutralizing response through vaccination, and now that we have achieved this we can optimize the vaccination strategy to something more feasible in real world conditions.

We are currently working on multiple approaches to improve the vaccine response speed and frequency. Relevant to this, we tested both GPC trimers and GPC trimer nanoparticles in rhesus macaques (Supplemental Fig. 3) and observed - by the third immunization – one animal with neutralizing titers in the trimer group and two animals with neutralizing titers in the GPC trimer-nanoparticle group. These studies suggest that neutralizing responses can be elicited with just three immunizations, though with only 20-30% consistently.

The nanobody D5 reveals a novel epitope, and this is of interest. However, the potency of the nanobody appears to be low even after making it a bispecific likely precluding it from being applied as an immunotherapeutic.

Response:

We agree with this reviewer that D5 has low potency and is unlikely to be used alone as an immunotherapeutic. We are currently working to improve D5 through multiple strategies.

Minor point

Line 92 Sanders needs a reference – should be Brouwer and colleagues. –Also, this information is redundant with line 86 – please edit.

Response:

We thank this reviewer for the helpful suggestions. We revised the text to “Brouwer et al.¹⁸ have shown LASV GPC-I53-50 nanoparticles...”

Reviewer #2 (Remarks to the Author):

The manuscript by Gorman et al. demonstrates that a prefusion-stabilized LASV GPC trimer was an improved antigen to elicit neutralizing antibodies, like GP23. A single domain antibody, D5, was found to recognize a glycan-free hole at the trimer apex and an asymmetrical cleavage intermediate of the GPC trimer is required to expose the “site-of-vulnerability”. In general, the manuscript has been prepared with high quality. I do have several comments that may improve the presentation.

Response:

We appreciate this reviewer’s summary of our study and the positive comment of our research being of high quality.

Major points:

Cryo-EM studies of D5-GPC complexes were carried out by using stabilized GPC trimers. Unlike this engineered version, native GPC trimers may be able to interact with D4(5) in a more flexible manner and incomplete cleavage between GP1 and GP2 could be unnecessary. Unless there is more direct evidence, it would be better to modify the text in regarding of asymmetrical cleavage of GPC on the viral surface.

Response:

We agree with the reviewer that another hypothesis for how D5 neutralizes native GPC on the viral surface could be through flexibility of the GP1 domains and “breathing” of the trimer rather than incomplete cleavage. We noted this in the discussion originally by stating “Native trimer instability may play a role in exposing this vulnerable conformation and/or state on the virion surface.” We have now expanded this, and we note the lack of direct evidence for the cleavage intermediate in the discussion as well.

Minor points:

1) In Fig. 1f, all error bars look the same. Are they correct?

Response:

We confirm that error bars are correct and note that the error bar on 50 degree is smaller than others.

3) In Fig. 2a, the labels for protein bands on SDS-PAGE are misplaced.

Response:

We thank this reviewer for pointing out the misplaced labels. We modified the figure; now the labels for protein bands are updated.

4) in Fig. 3d, the colors for the 18.5C heavy and light chain are indistinguishable.

Response:

We have now adjusted the figure to color the chains differently so they may be distinguished. We note that the antibody label has been corrected as well.

5) in Fig. 5, the GPC part shows high resolution, while the electron density for antibodies is relatively unclear. Is the resolution high enough for model building of antibodies?

Response:

The density for the antibody interactions is quite clear, however it is hard to contour the full structure density to show this. We have now added a panel to show the quality of the antibody density to the readers.

Reviewer #3 (Remarks to the Author):

In the manuscript entitled “Cleavage-Intermediate Lassa Virus Trimer Elicits Neutralizing Responses, Identifies Neutralizing Nanobodies, and Reveals an Apex-Situated Site-of-Vulnerability”, the authors engineered the glycoprotein complex (GPC) of Lassa virus based on reported structures and profound antigenic screening. The authors then characterized the engineered GPC trimer with known antibodies and demonstrated that the GPC construct in the nanoparticle format could elicit neutralizing antibodies in guinea pigs, and have successfully identified the first guinea pig-derived anti-LASV neutralizing mAb (GP23). Further, one potent neutralizing nanobody (D5) was sorted out by using the GPC construct. The aim of the project, which is the rational design of a suitable GPC construct for both immunization and neutralizing antibody discovery, is of great interest in the field, because the neutralizing antibody response is protective but extremely rare for Lassa. However, there are a couple of conflicts between the structural and biochemical results of the manuscript, as well as with previous studies, that need to be solved.

Response:

We appreciate this reviewer’s in-depth summary of our study and the positive comments. We address this reviewer’s comments in full as detailed below.

Major comments:

1. Page 4: in the paragraph describing the history of LASV GPC antigen design and production, one well-designed LASV GPC trimer, named GPCysRRLL (PMID: 36288283), was missed. This GPC antigen is a stabilized prefusion trimer with the native RRLC cleavage motif for the LASV receptor (matriglycan) recognition. Importantly, it is the only reported LASV GPC construct that is able to be recognized by anti-LASV human neutralizing antibodies from all four discovered human neutralizing epitope groups (GP1-A, GPC-A, GPC-B, and GPC-C), including 8.9F, an important preclinical therapeutic mAb which binds to the apex of GPC and directly blocks GPC/matriglycan receptor binding. Since one of the main features of this manuscript is GPC immunogen design for neutralizing antibody elicitation, it is recommended to include and discuss the GPCysRRLL construct in the text and table.S1.

Response:

Thank you for pointing out the missing citation for GPCysRRLL. We included this design in Supplementary Table 1. We also revised the introduction as shown below.

“This construct, GPCysR4, was modified to substitute the native SKI-1 cleavage site (RRLC) with a site amenable to furin cleavage. In addition, Saphire and colleagues also created another construct, GPCysRRLL, as it has been shown that the RRLC cleavage site is important for receptor binding¹⁵ and recognition of neutralizing antibody 8.9F¹⁷.”

2. Line 171: “...but gradually rose so that by the 5th and 6th nanoparticle immunization, they averaged about 500 IC50.” The neutralizing titers of the immunized guinea pig sera are the crucial results of this manuscript. The measurement unit of the neutralizing titer is shown as “IC50”, however, the authors didn’t identify the “IC50” here. Is it from a specific monoclonal antibody or a serum sample? In addition, since there was one animal showed a much higher titer than others, it is recommended to use a median titer than an average titer. Plus, a typo in Fig. 2e: “ID50”.

Response:

We thank this reviewer for correcting the usage of ID50 and IC50. In Figure 2e, we tested the serum samples, therefore, we revise the “IC50” to “ID50” in the text. In the method section on page 24, we defined the ID50/IC50. We also report the median in addition to the average as the reviewer suggests.

“Neutralization titers (ID50: 50% inhibitory dose or IC50: 50% inhibitory concentration) were defined as the serologic sample dilution or antibody concentration at which the relative light units (RLUs) were

reduced by 50% compared with the virus control wells (virus + cells) after subtraction of the background RLUs in the control groups with cells only.”

3. Line 204: “Binding competition revealed that nanobodies C3 and D5 competed for binding to GPC with known human LASV neutralizing antibodies from the GPC-B group”. The epitope of the nanobody D5 binds to the apex of GPC trimer (Fig.5), which is far always from the epitope of GPC-B mAbs. Could the authors explain why D5 and GPC-B mAbs compete with each other?

Response:

We show that the structure revealed allosteric inhibition of apex binding of D5 on the recognition of GPC-B antibody. We explained the details on page 11, as show below.

“The structure of the trimer bound by D5 displayed an asymmetric assembly and when compared to crystallized GPCs^{11,16}, a single protomer aligned with a root-mean-square deviation (rmsd) of 1.5 Å after removing outliers; however, neighboring protomers extended over 17 Å farther at certain regions near the apex with overall rmsds of 13.2 Å and 21.9 Å (Fig. 6a). This extension resulted in the allosteric disruption of 37.7H binding, and stabilization of this conformation by D5 appeared to prevent 37.7H from binding, thereby providing an explanation for the observed competition between D5 and GPC-B antibodies, which bind across adjacent GPC protomers distal from the apex (Fig. 6b).”

4. Fig.5d: The structure models show the epitopes of 18.5C and 8.11G are overlapped, however, no binding competition is shown in Fig.S6C. In fact, 18.5C (GPC-B) and 8.11G (GPC-A) belong to two distinct epitope groups (PMID: 27161536).

Response:

We thank the reviewer for catching this. This antibody should have been 36.1F and has now been relabeled correctly. The confusion stemmed from an exact technical transposition in the PDB deposit 7S8H with both antibodies in complex with GPC. The authors had labeled 36.1F with chains B+C and 18.5C with H+L and the PDB then relabeled the chains of 18.5C as B+C and 36.1F to H+L so the PDB web sequence display do not match in labels. This has now been corrected.

5. About the neutralizing mechanism of nanobody D5: the authors found D5 only recognized the cleavage-intermediate GPC trimer but not the mature, fully cleaved trimer, and suggested it neutralized LASV by blocking receptor matriglycan. However, based on the previous structural study (PMID: 35173332), the binding of GPC/matriglycan required a mature GPC trimer with fully-cleaved RRLL motifs of all three protomers. Therefore, if D5 cannot recognize mature GPC, it wouldn't block the matriglycan binding for neutralization. On the other hand, even if the matriglycan is able to bind to the cleavage-intermediate GPC for virion cell entry, the uncleaved GP1/GP2 protomer(s) in the cleavage-intermediate GPC is not able to mediate virus-endosome membrane fusion. The author may dig deep to solve the issue on D5 nanobody. In addition, the “site-of-vulnerability” on the apex of the trimer has been discovered in the previous study (PMID: 36288283). It is not a new site.

Response:

Our discussion point regarding the steric occlusion of the matriglycan and D5 binding a mature GPC was potentially unclear. It followed just after the premise that instead of incomplete cleavage D5 could bind to a mature GPC that had flexibility in the GP1 apex, such “breathing” is observed for HIV-1 Env on virions. In that case it would bind, occlude, and disturb the conformation of the site for matriglycan binding. When we show D5 would not be compatible with mature GPC, that is in the context of what has been observed in crystal structures or cryoEM stabilized conformations, not what may be observed in more dynamic environment on the surface of a virion. For the novelty of the site, we have added some clarification to distinguish it from the 8.9F epitope. Supplemental Figure 7e did not do a sufficient job illustrating the difference between he 8.9F site and the D5 site, we have now expanded this to highlight that D5 is central to the apex, whereas 8.9F is on the periphery of the apex, there is little overlap.

6. For further evaluation, it is required to show the formal, complete PDB validation reports of the EM-maps and atomic models along with the manuscript.

Response:

Complete PDB validation reports of the EM-maps and atomic models were added in the supplemental material.

Minor comments:

1. Page 5, line 89: “The highly glycosylated GPC induces weak and inconsistent immune responses in both natural infection and vaccination”. Indeed, both T and B cell responses are potent after infection or immunization, however, just neutralizing antibody response is extremely weak and inconsistent.

Response:

We appreciate this reviewer’s comments. Now we edit the text to

“Although highly glycosylated GPC can induce potent T and B cell immune responses, the development of neutralizing antibodies in response to GPC is often weak and inconsistent.”

2. Could the authors describe how many copies of GPC trimer are displayed on each encapsulating nanoparticle?

Response:

The Encapsulin nanoparticle has the capability to attach a maximum of 20 copies of GPC trimer onto its surface. From the negative EM staining, a minimum of 10 copies of GPC trimer can be identified. Therefore, it is estimated that the nanoparticle surface can display between 10 to 20 copies of Lassa GPC trimer. We now state this in the methods.

3. Fig. 3d: Could the authors show the side view of the EM-map?

Response:

Yes, we now display the side view in Figure 3D.

4. Fig. S3a and b: What does the MF4 mean here?

Response:

MF4 is an anti-Foldon specific antibody. Foldon is used in our design to stabilize Lassa GPC trimer conformation. Immunization with GPC trimer will elicit anti-Foldon responses. Here we used MF4 antibody as a control. We now state this in the methods and in the figure legend.

5. Fig. 7d: Please display or write which colors represent the “Conserved” and “Variable” regions.

Response:

In Fig.7d, we do have a color bar below the figure 7D. We have moved the labels to the sides to make it clearer.

6. Typos:

Fig.2e and 3a: “ID50 neutralization”

Table S1: “G326C”, should it be L326C?

Response:

Thank you for pointing out the typos. We correct it “G326C” to “L326C”.

REVIEWER COMMENTS

Reviewer #1 (Remarks to the Author):

The authors have addressed my comments appropriately.

Reviewer #2 (Remarks to the Author):

The authors have adequately addressed the issues I raised. I support the publication of the revised paper.

Reviewer #3 (Remarks to the Author):

The authors have addressed the previous questions well. Additional points may be addressed before publication:

1. According to the PDB validation report and Table 1, the Q-score and EMRinger Score of the GPC/D5/8.11G complex are low, which indicate the overall quality of the EM density map. Could the authors display the entire density details of both GPC/D5 and GPC/8.11G interfaces? The view shown in Fig.5b (right panel) is too narrow.
2. In Table 1, the information of the GPC/GP23-Fab complex is missed.
3. In Supplementary Table 1, the GPCysRRL construct had a trimerization domain (named 1NOG) and was expressed from HEK 293F cells.

Point-by-Point Response to Reviewers

Reviewer #1 (Remarks to the Author):

The authors have addressed my comments appropriately.

Response:

We are delighted to hear that we had already addressed all the issues from this reviewer.

Reviewer #2 (Remarks to the Author):

The authors have adequately addressed the issues I raised. I support the publication of the revised paper.

Response:

We are delighted to hear that we had already addressed all the issues raised by reviewer #2.

Reviewer #3 (Remarks to the Author):

The authors have addressed the previous questions well. Additional points may be addressed before publication:

Response:

We are delighted to hear that we had addressed the previous questions from this reviewer. We have addressed the additional questions from this reviewer as detailed below.

1. According to the PDB validation report and Table 1, the Q-score and EMRinger Score of the GPC/D5/8.11G complex are low, which indicate the overall quality of the EM density map. Could the authors display the entire density details of both GPC/D5 and GPC/8.11G interfaces? The view shown in Fig.5b (right panel) is too narrow.

Response:

We agree with the reviewer that we should show more density examples to give readers an impression of the overall quality of the map and we now have expanded on the density panel we added in the last revision to add two more to a main figure. Figure 5 now shows the quality of the density at the engineered disulfide, at the antibody interface, and at the nanobody interface. We would however note that the Q-score is not low for a 4.7 Å structure, according to EMDB statistics for maps in the 4-5 Å range, an overall Q-score of 0.3 is right at the peak of the distribution (see figure below for EMDB statistics). We also feel the EMRinger score of 0.97 is again, reasonable for a 4.7 Å resolution with scores in the EMRinger article far below ours for this resolution range. Though we concede refinement methods have improved since it was initially published and we don't have updated statistics on more recent structures, we note that 1.0 was a goal pointed out in the manuscript for much higher resolution structures than ours. "An EMRinger score of 1.0 sets an initial quality goal for a model refined against a map in the 3.2–3.5Å range, while very high quality models at high resolution generate scores above 2.0." -Barad et al, Nat Methods. 2015.

2. In Table 1, the information of the GPC/GP23-Fab complex is missed.

Response:

We thank the reviewers for pointing this out. We have now added to Table 1 the information for this map and deposited it in EMDB.

3. In Supplementary Table 1, the GPCysRRLL construct had a trimerization domain (named 1NOG) and was expressed from HEK 293F cells.

Response:

We thank this reviewer for pointing out the mistakes. We have updated and revised Supplementary Table 1.

REVIEWERS' COMMENTS

Reviewer #3 (Remarks to the Author):

The authors have addressed the comments. No more concerns.

Point-by-Point Response to Reviewers

Reviewer #1 (Remarks to the Author):

The authors have addressed my comments appropriately.

Response:

We are delighted to hear that we had already addressed all the issues from this reviewer.

Reviewer #2 (Remarks to the Author):

The authors have adequately addressed the issues I raised. I support the publication of the revised paper.

Response:

We are delighted to hear that we had already addressed all the issues raised by reviewer #2.

Reviewer #3 (Remarks to the Author):

The authors have addressed the previous questions well. Additional points may be addressed before publication:

Response:

We are delighted to hear that we had addressed the previous questions from this reviewer. We have addressed the additional questions from this reviewer as detailed below.

1. According to the PDB validation report and Table 1, the Q-score and EMRinger Score of the GPC/D5/8.11G complex are low, which indicate the overall quality of the EM density map. Could the authors display the entire density details of both GPC/D5 and GPC/8.11G interfaces? The view shown in Fig.5b (right panel) is too narrow.

Response:

We agree with the reviewer that we should show more density examples to give readers an impression of the overall quality of the map and we now have expanded on the density panel we added in the last revision to add two more to a main figure. Figure 5 now shows the quality of the density at the engineered disulfide, at the antibody interface, and at the nanobody interface. We would however note that the Q-score is not low for a 4.7 Å structure, according to EMDB statistics for maps in the 4-5 Å range, an overall Q-score of 0.3 is right at the peak of the distribution (see figure below for EMDB statistics). We also feel the EMRinger score of 0.97 is again, reasonable for a 4.7 Å resolution with scores in the EMRinger article far below ours for this resolution range. Though we concede refinement methods have improved since it was initially published and we don't have updated statistics on more recent structures, we note that 1.0 was a goal pointed out in the manuscript for much higher resolution structures than ours. "An EMRinger score of 1.0 sets an initial quality goal for a model refined against a map in the 3.2–3.5Å range, while very high quality models at high resolution generate scores above 2.0." -Barad et al, Nat Methods. 2015.

2. In Table 1, the information of the GPC/GP23-Fab complex is missed.

Response:

We thank the reviewers for pointing this out. We have now added to Table 1 the information for this map and deposited it in EMDB.

3. In Supplementary Table 1, the GPCysRRLL construct had a trimerization domain (named 1NOG) and was expressed from HEK 293F cells.

Response:

We thank this reviewer for pointing out the mistakes. We have updated and revised Supplementary Table 1.